# Can Diffusion Models Learn Hidden Inter-Feature Rules Behind Images?

**Yujin Han** [* 1]  **Andi Han** [* 2 3]  **Wei Huang** [3]  **Chaochao Lu** [4]  **Difan Zou** [1]

## Abstract

Despite the remarkable success of diffusion models (DMs) in data generation, they exhibit specific failure cases with unsatisfactory outputs. We focus on one such limitation: the ability of DMs to learn hidden rules between image features. Specifically, for image data with dependent features ($\mathbf{x}$) and ($\mathbf{y}$) (e.g., the height of the sun ($\mathbf{x}$) and the length of the shadow ($\mathbf{y}$)), we investigate whether DMs can accurately capture the inter-feature rule ($p(\mathbf{y}|\mathbf{x})$). Empirical evaluations on mainstream DMs (e.g., Stable Diffusion 3.5) reveal consistent failures, such as inconsistent lighting-shadow relationships and mismatched object-mirror reflections. Inspired by these findings, we design four synthetic tasks with strongly correlated features to assess DMs' rule-learning abilities. Extensive experiments show that while DMs can identify coarse-grained rules, they struggle with fine-grained ones. Our theoretical analysis demonstrates that DMs trained via denoising score matching (DSM) exhibit constant errors in learning hidden rules, as the DSM objective is not compatible with rule conformity. To mitigate this, we introduce a common technique - incorporating additional classifier guidance during sampling, which achieves (limited) improvements. Our analysis reveals that the subtle signals of fine-grained rules are challenging for the classifier to capture, providing insights for future exploration.

## 1. Introduction

Despite the remarkable capabilities demonstrated by diffusion models (DMs) in generating realistic images (Ho et al., 2020; Song et al., 2020; Vahdat et al., 2021; Dhariwal

---
[*]Equal contribution  [1]The University of Hong Kong [2]The University of Sydney  [3]RIKEN AIP  [4]Shanghai AI Laboratory.  Correspondence to:  Yujin Han <yujin-han@connect.hku.hk>, Andi Han <andi.han@sydney.edu.au>, Difan Zou <dzou@cs.hku.hk>.

*Proceedings of the $42^{nd}$ International Conference on Machine Learning*, Vancouver, Canada. PMLR 267, 2025. Copyright 2025 by the author(s).

& Nichol, 2021; Karras et al., 2022; Tian et al., 2024b), videos (Ho et al., 2022; Yu et al., 2024; Yuan et al., 2024b), and audio (Liu et al., 2023a; Yang et al., 2024b; Lemercier et al., 2024), they still encounter specific failures in synthesis quality, such as anatomically incorrect human poses (Borji, 2023; Zhang et al., 2024; Huang et al., 2024) and misalignment between generated content and prompts (Feng et al., 2022; Borji, 2023; Chefer et al., 2023; Liu et al., 2023b; Lim & Shim, 2024), which could harm the reliability and applicability of DMs in real-world scenarios.

We focus on a specific type of failure with limited attention: the failure of DMs in learning hidden inter-feature rules behind images. Specifically, consider image data containing dependent feature pairs ($\mathbf{x}, \mathbf{y}$), such as the height of the sun ($\mathbf{x}$) affecting the length of a pole's shadow ($\mathbf{y}$). Our investigation centers on whether DMs targeting the joint distribution $p(\mathbf{x}, \mathbf{y})$ can accurately capture the underlying relationships between $\mathbf{x}$ and $\mathbf{y}$, effectively recovering the conditional distribution $p(\mathbf{y}|\mathbf{x})$. Theoretically, a diffusion model that perfectly estimates the joint distribution should naturally capture the conditional distribution, thereby learning the latent rules between features. However, in practice, numerous factors, such as non-negligible score function estimation errors, can cause the sampled joint distribution to deviate significantly from the true distribution (Chen et al., 2022; 2023; Benton et al., 2024). How do these deviations propagate to inter-feature rule learning? This gap between theory and practice remains largely unexplored.

Although existing studies have explored whether DMs can learn specific rules, they primarily focus on independent features, such as DMs' compositional capabilities (Okawa et al., 2024; Deschenaux et al., 2024; Wiedemer et al., 2024). Some works have investigated inter-feature dependencies in DMs, but the varying complexity of rules has led to contradictory findings. For example, DDPM has been reported to fail in generating images satisfying numerical equality constraints (Anonymous, 2025), while succeeding in reasoning about shape patterns in RAVEN task (Wang et al., 2024a). These inconsistencies highlight the need for a unified experimental setting that allows for adjustable rule difficulty, enabling an accurate evaluation of DMs' rule-learning capabilities. Moreover, existing studies rely heavily on empirical observations, lacking theoretical analysis to elucidate the limitations of DMs in rule conformity.

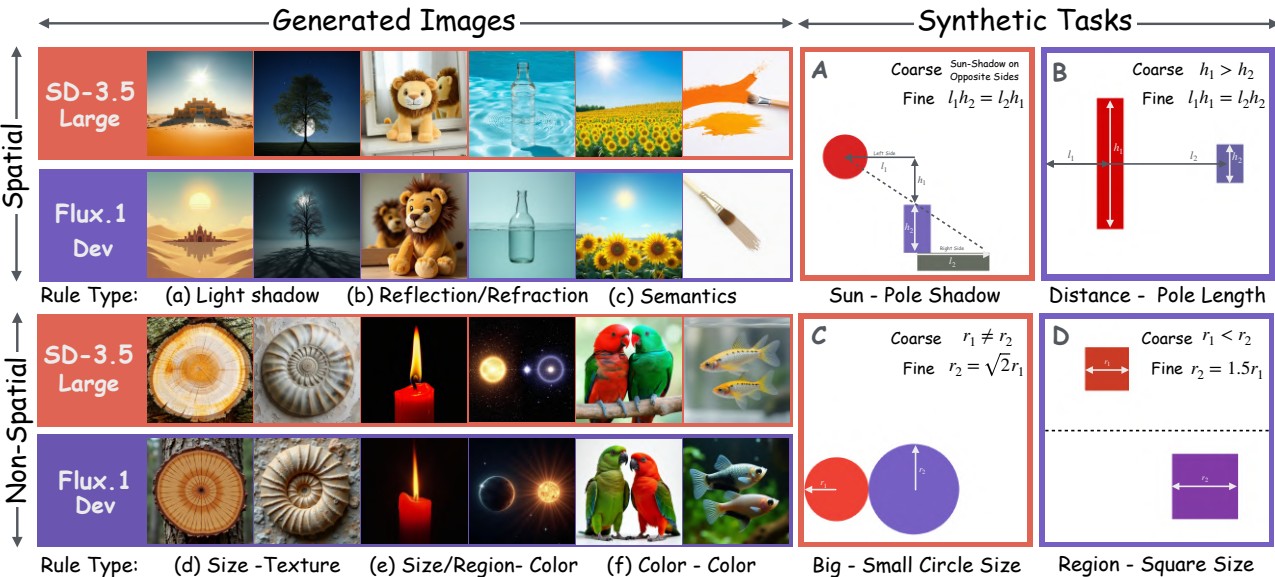

Figure 1. **Synthetic Tasks Inspired by Real-World Insights.** Based on whether inter-feature rules involve spatial dependencies, we categorize the failure cases into spatial and non-spatial rules. **Spatial rules** include: (a) Light-shadow, where evaluated DMs generate unreasonable multiple shadows or incorrect shadow flips; (b) Reflection/Refraction, showing incorrect mirror rules or missing refraction effects below water surface; (c) Semantics, such as inconsistencies between sunflower orientation and sun position, or brush and canvas colors. **Non-spatial rules** involve: (d) Size-Texture, like mismatches between tree diameter and growth rings; (e) Size/Region-Color, where evaluated models fail to capture burning candle's color variations and star size-color relationships (e.g., red giants and white dwarf); (f) Color-Color, as in Eclectus parrots' body-beak color correlations that DMs fail to maintain. Appendix C provides detailed explanations for each case. These failures of mainstream DMs in handling real-world inter-feature rules inspire our design of four synthetic tasks.

Our investigation into inter-feature rules begins with observing the limited ability of mainstream DMs (e.g., `SD-3.5 Large`, `Flux.1 Dev`) to capture real-world inter-feature rules, as illustrated in Figure 1, even though these models perform well on metrics such as FID [1]. Their errors in inter-feature relationships are evident in various scenarios, such as inconsistent relationships between sun positions and building shadows, mismatched reflections of toys in mirrors, and sunflowers failing to face the sun. Then, we carefully design four synthetic tasks to reflect real-world rule failures, ensuring the practical relevance of our findings. The rule of each task features two difficulty levels: coarse-grained rules (e.g., the sun and a pole's shadow should be on opposite sides) and fine-grained rules (e.g., the shadow's length as a precise function of the sun's height). This hierarchical, controllable framework enables a comprehensive evaluation of DMs' rule-learning capabilities. Next, through extensive experiments considering various factors including model architectures, training data size, and image resolution, we reach a consistent conclusion: *DMs effectively learn coarse-grained rules but struggle with fine-grained ones.*

Furthermore, we develop a rigorous theoretical analysis using a multi-patch data model with an inter-feature rule

specified in terms of norm. We prove a constant error lower bound on learning the hidden rule via optimizing the DSM objective (Ho et al., 2020) with a two-layer network. This demonstrates the incompatibility between learning joint distributions and identifying specific inter-feature rules.

Recognizing DMs' difficulty in learning inter-feature rules, we mitigate this issue by constructing contrastive pairs that satisfy either fine-grained or coarse-grained rules and then using them to train a classifier as additional guidance. While this strategy enhances rule-compliant sample generation, further improvements are still achievable. The in-depth analysis identifies that fine-grained rules exhibit weak signals, making accurate classifier training particularly challenging. We summarize our **key contributions** as follows:

*Empirically*, inspired by mainstream DMs' struggles with real-world inter-feature rules, we innovatively create synthetic tasks with coarse/fine-grained rules to systematically assess DMs' rule learning ability in Section 3.1.1. Extensive experiments in Section 3.3 show that while DMs can learn coarse rules, their ability to grasp precise rules is limited.

*Theoretically*, we rigorously analyze DMs on a synthetic multi-patch data distribution with a hidden norm dependency in Section 4. We prove that the unconditional DDPM cannot learn the precise rule of norm constraint, which exhibits at least a constant error in approximating the desired

---

[1]Appendix A lists Mixture Gaussian as an example to demonstrate that low FID and incorrect inter-feature relationships in DMs' generations are not contradictory.

score function. This identifies the limitation of the current DMs training paradigm and necessitates further improvements for learning hidden rules behind images.

*Methodologically*, we mitigate DMs' inability to learn fine-grained rules by introducing guided diffusion with a contrastive-trained classifier in Section 5. However, the challenges of accurately classifying fine-grained rules identify room for improvement in our strategy. This problem, distinct from traditional classification tasks, involves detecting subtle distinctions between fine-grained and coarse-grained rules, highlighting valuable insights for future exploration.

## 2. Related Work

We summarize prior studies on the ability of DMs to learn specific rules, and discuss the relations to inter-feature rules.

**Factual Knowledge Rules.** The violation of factual rules in DMs refers to generated images failing to accurately reflect factual information and common sense, often characterized as hallucinations in existing work (Aithal et al., 2024; Lim & Shim, 2024; Anonymous, 2025). Typical examples include violating common sense, such as extra, missing, or distorted fingers (Aithal et al., 2024; Pelykh et al., 2024; Ye et al., 2023), unreadable text (Gong et al., 2022; Tang et al., 2023; Xu et al., 2024) and snowy deserts (Lim & Shim, 2024). Additionally, inconsistencies between generated images and given textual prompts (Liu et al., 2023b; Fu & Cheng, 2024; Mahajan et al., 2024; Li et al., 2024b) can be regarded as violations of prompt-based knowledge. Unlike inter-feature rules, factual knowledge rules *do not involve relationships between multiple features* and are typically attributed to imbalanced training data distribution (Samuel et al., 2024) or mode interpolation caused by inappropriate smoothing of training data (Aithal et al., 2024).

**Independent Features Rules.** Prior work has investigated DMs' ability to combine independent features, i.e., compositionality. Through controlled studies with independent concepts (e.g., color, shape, size), Okawa et al. (2024) observe that DDPM can successfully compose different independent features. Similar findings are reported in (Deschenaux et al., 2024), where interpolation between portraits without and with clear smiles resulted in generations with mild smiles. However, numerous studies highlight DMs' limitations in complex compositional tasks (Liu et al., 2022; Gokhale et al., 2022; Feng et al., 2022; Marioriyad et al., 2024), potentially due to insufficient training data for reconstructing each individual feature (Wiedemer et al., 2024). These studies primarily examine compositional tasks with *independent features*, in contrast to our focus on feature dependencies.

**Abstract (Dependent Feature) Rules.** This type closely aligns with our work, which studies feature relationships like shape consistency in generations. Prior studies give mixed

conclusions on DDPM's rule-learning ability. For example, DDPM struggles with numerical addition rule (Anonymous, 2025) but maintains shape consistency rule in RAVEN task (Wang et al., 2024a). Inconsistent rule complexity leads to ambiguous evaluation conclusions, and the lack of theoretical analysis leaves the underlying factors behind DMs' performance in rule learning poorly understood. Through controlled experiments with adjustable rule complexity, we provide a unified assessment of DMs' rule-learning abilities and offer a theoretical explanation of their fundamental limitations, as a result of their training paradigm.

## 3. Exploring Hidden Inter-feature Rule Learning via Synthetic Tasks

In real-world image generation tasks, rules between features are often complex and difficult to define or quantify precisely. To systematically investigate DMs' ability in rule learning, as previous work (Okawa et al., 2024; Deschenaux et al., 2024; Anonymous, 2025; Wang et al., 2024a), we design simplified and controllable synthetic tasks in Figure 1. These synthetic tasks not only provide explicitly defined inter-feature rules but also abstract essential feature rules present in real-world data, thereby making our conclusions practically relevant. For example, Synthetic Task A in Figure 1 simulates the *Light-Shadow* relationship, while Task B simplifies the physical rules of *Reflection/Refraction*.

### 3.1. Synthetic Tasks Inspired by Real-World Insights

#### 3.1.1. REAL-WORLD HIDDEN INTER-FEATURE RULES

Inspired by Borji (2023), we investigate several common scenarios where inter-feature rules exist, as illustrated in Figure 1. Specially, we categorize these hidden rules into two types, *spatial rules* and *non-spatial rules*, based on whether the inter-feature relationships exist in the form of spatial arrangements or feature attributes themselves.

**Spatial Rules** are defined as constraints on the relative positions and layouts between features, such as the correlation between the sun's height and the shadow's length. In Figure 1, scenario *Light-shadow* demonstrates how the position of a light source should precisely determine the placement of building shadows. However, both 8-billion Multimodal SD-3.5 Large[2](Rombach et al., 2022) and 12-billion model Flux.1 Dev[3](Labs, 2023), fail to generate proper shadows, either producing incorrect directions or merely creating symmetrical duplicates of the actual buildings. Similarly, in scenario *Reflection/Refraction*, while objects in front of mirrors should dictate the layout of their reflections, we observe completely unreasonable generations from both

---

[2]https://huggingface.co/spaces/stabilityai/stable-diffusion-3.5-large

[3]https://fal.ai/models/fal-ai/flux/dev

models. Furthermore, semantic consistency in *Semantics* scenario is violated, as shown by sunflowers not facing the sun and mismatched paint colors between brush and canvas.

**Non-Spatial Rules** are defined as correlations between intrinsic feature attributes, such as the relationship between an object's size and its color. For instance, in type *Size -Texture*, tree trunk features should exhibit precise correlations between the diameter and annual ring count, and candle flames in type *Size/Region- Color* should show constrained relationships between different flame zones and their colors. However, these fine-grained inter-feature constraints are ignored by both `SD-3.5 Large` and `Flux.1 Dev`. More detailed discussion and additional experiments for more advanced DMs are deferred to Appendix C.

### 3.1.2. SYNTHETIC TASKS

Inspired by real-world rules in Section 3.1.1, we design four synthetic tasks (A-D), each with two levels of rule granularity (coarse and fine), as shown in Figure 1. We provide a brief overview of synthetic tasks here, with more details presented in Appendix D. Specially,

**Task A** is inspired by the spatial rules behind the *Light-shadow* case, simulating the physical law between the sun and pole shadows. In Task A of Figure 1, the *coarse-grained rule* requires the sun and shadow to be on opposite sides of the pole, while the *fine-grained rule* requires sun's center, pole top, and shadow endpoint align linearly, i.e., satisfying $l_1 h_2 = l_2 h_1$ (see notations in Task A, Figure 1).

**Task B** abstracts the spatial rule from the *Reflection/Refraction* case, where an object's reflection size depends on its size and distance from the mirror. Task B uses two rectangles with lengths $h_1$ and $h_2$ (notations shown in Task B, Figure 1) to simulate this perspective rule, where size diminishes with distance. Assuming the viewpoint is at the leftmost edge, the *coarse-grained rule* requires the left rectangle (closer to the viewpoint) to be longer than the right one (farther from the viewpoint), i.e., $h_1 > h_2$, while the *fine-grained rule* dictates rectangle lengths be proportional to their distances from the viewpoint, i.e., $l_1 h_2 = l_2 h_1$.

**Task C** consists of two tangent circles of different radii, aiming to capture the relationship between shape/outlook and size as illustrated in non-spatial rule. The *coarse-grained rule* simply requires distinct radii for the two circles, i.e., $r_1 \neq r_2$, while the *fine-grained rule* specifies a precise ratio between the radii, requiring $r_2 = \sqrt{2} r_1$.

**Task D** simplifies the non-spatial rule from scenario *Size/Region- Color* in Figure 1, where, in candle flame generations, colors transition from blue near the wick to yellow at the outer regions. We construct two such squares, with smaller squares positioned in the upper half and larger ones in the lower half of the image. The *coarse-grained rule*

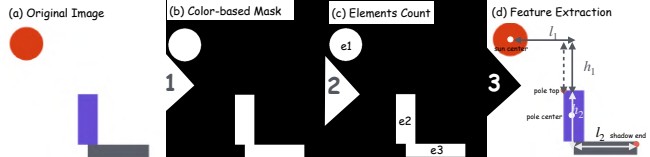

*Figure 2.* **Pipeline for extracting features.** Given an image, we first apply a color-based mask by using predefined colors, then count whether the number of masks meets expectations, and finally extract features of interest by marking the key points within masks.

requires that the upper square's side length $l_1$ be smaller than the lower square's side length $l_2$, i.e., $l_1 < l_2$, while the *fine-grained rule* specifically requires $l_2 = 1.5l_1$.

### 3.2. Experimental and Evaluation Setup

After designing synthetic tasks with well-defined inter-feature rules, we can systematically investigate the capability of DMs to learn these underlying relationships.

**Experimental Setup.** In subsequent experiments, we train an unconditional DDPM (Ho et al., 2020) on four synthetic tasks. Unlike latent-space DMs (e.g., `SD-3.5 Large`), pixel-space DDPM makes the conformity of inter-feature relationships potentially simpler, as no additional compression-induced information loss occurs (Rombach et al., 2022; Yao & Wang, 2025). Following the training setting (Aithal et al., 2024), we fix the total timesteps at $T = 1000$ and employ the widely-used U-Net architecture (Ronneberger et al., 2015) as the denoiser. 4000, 2000, 2000, and 2000 samples are generated for synthetic task A, B, C and D, respectively, with an image size of $32 \times 32$. Additionally, in Appendix E.3, we explore more advanced architectures such as DiT (Peebles & Xie, 2023) and SiT (Ma et al., 2024), alongside larger synthetic datasets of 20000 and 40000 samples and higher image resolutions of $64 \times 64$. These factors enhance the training of DMs, thus leading to better alignment between generated and real data distributions (Chen et al., 2022; Benton et al., 2024; Chen et al., 2023) and enabling more effective learning of hidden rules. More experimental details are in Appendix E.1.

**Evaluation Method.** To evaluate whether generated images follow the inter-feature rules, Figure 2 designs a three-step feature extraction pipeline: (1) Color-based Mask: Segment element masks (e.g., sun, pole, shadow in Task A) based on predefined color (HSV) ranges when synthesizing training data; (2) Elements Count: Apply contour detection based on masks to verify the presence of essential elements, marking images as `Invalid` if any are missing; (3) Feature Extraction: Extract key feature points (e.g., sun center, pole top/center and shadow endpoint in Figure 2) and compute geometric features of interest, such as horizontal sun-to-pole distance $l_1$, vertical sun-to-pole-top distance $h_1$, pole height $h_2$, shadow length $l_2$. All features are scaled to $[0, 1]$ by dividing them by the image size to eliminate scale effects.

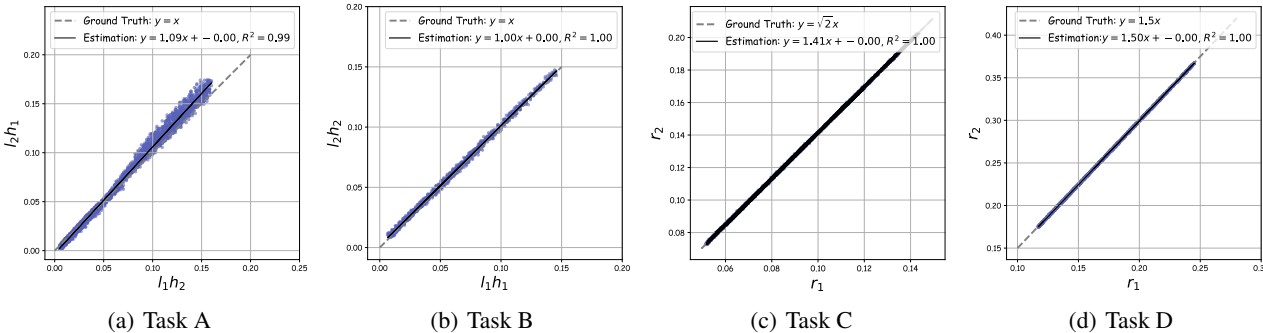

*Figure 3.* **Synthetic training data satisfies fine-grained rules.** To validate the evaluation method, we extract relevant features from the synthetic training data and check if they meet expectations, focusing on generations within the interval $[2.5\%, 97.5\%]$ for stability. The closely matching Estimation and Ground Truth lines, along with an $R^2$ value near 1, demonstrate effectiveness of the evaluation method.

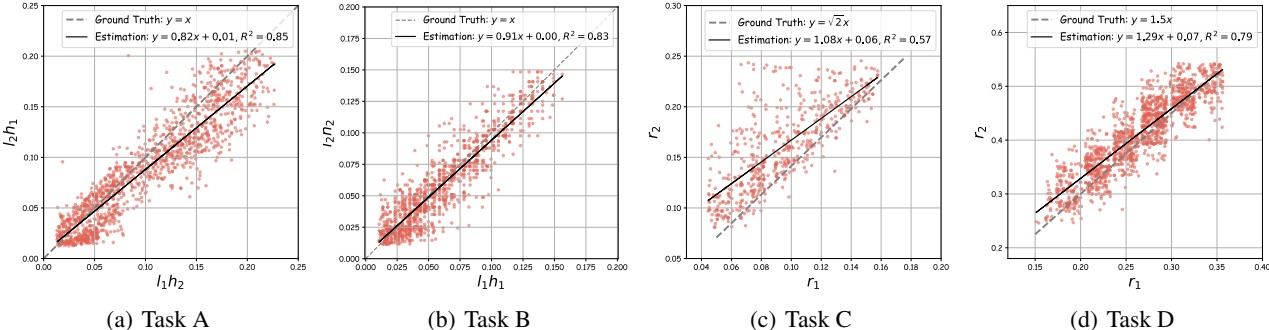

*Figure 4.* **Generated data does not satisfy fine-grained rules.** Considering generated samples within the $[2.5\%, 97.5\%]$ range, we extract focused features and check if they meet fine-grained rules. The Estimation line, far from the Ground Truth line, and an $R^2$ value less than 1, reveal DMs' failure in learning fine-grained rules. Appendix E.2 shows generated images that violate the fine-grained rules.

With these features, we can verify whether generated images satisfy predefined rules. For example, in Task A, we examine: (1) Coarse-grained rule: the sun and shadow are on opposite sides of the pole by comparing the relative positions of the sun center, pole center, and shadow endpoint; (2) Fine-grained rule: validate the precise geometric relationship $l_1h_2 = l_2h_1$. We extend the same feature extraction approach in Figure 2 to validate inter-feature rules in Tasks $B$, $C$, and $D$. We apply the evaluation method to synthetic training data to validate our approach's effectiveness, as shown in Figure 3, which demonstrates a close alignment between the estimation and ground truth across all tasks.

### 3.3. Experimental Results

For each synthetic task, we generate 2000 samples and report the evaluated results as follows:

**DMs' Success on Coarse-Grained Rules.** Table 1 demonstrates that DMs rarely generate samples that violate the coarse-grained rules across all tasks. This observation aligns with expectations: generating samples that violate coarse-grained rules requires DMs to generate out of the (training) distribution (OOD) - an extrapolation challenge for DMs observed in prior work (Okawa et al., 2024; Kang et al., 2024). In Task A, for example, all training samples place

*Table 1.* **DMs satisfy coarse rules.** Table 1 shows the invalid ratio is around 20%–40%. And DMs can learn coarse rules with one exception in Task A, which is visualized in Appendix E.1.

| Task | Invalid (%) | Coarse-Grained Violations |
|------|-------------|---------------------------|
| A | 30.15 | 1 |
| B | 40.45 | 0 |
| C | 41.75 | 0 |
| D | 24.90 | 0 |

the sun and shadow on opposite sides of the pole; violating this rule would require generating a never-seen mode with both elements on the same side.

**DMs' Failure on Fine-Grained Rules.** While following coarse-grained rules only requires DMs to avoid unreasonable OOD generations, fine-grained rules are much harder, demanding accurate learning of the in-distribution training data. Figure 4 demonstrates the models' performance across four synthetic tasks, where deviations from the ground truth in linear fitting and the coefficient of determination $R^2$ below 1 indicate that DMs fail to fully capture the predefined fine-grained rules. Additionally, we observe that DMs struggle more with learning non-spatial rules, such as Task C, compared to spatial rules, such as Task A, as evidenced by worse linear fitting and smaller $R^2$. This discrepancy likely

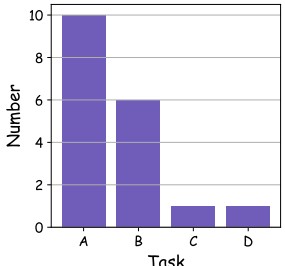
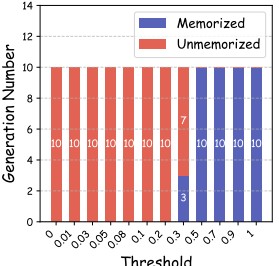

(a) Rule-conforming genera-
tions acrross four tasks.

(b) Memorization with differ-
ent thresholds in Task A.

*Figure 5.* **DMs generate rule-conforming samples.** Define Rule-conforming generations have ratios (e.g., $\frac{l_2 h_1}{l_1 h_2}$ in Task A) within $\pm 0.01$ of true ratio (1 in Task A). Figure 5(a) shows DDPM's ability to generate rule-conforming samples across tasks. Figure 5(b) indicates that nearest neighbor distances between 10 rule-conforming samples in Task A and training data are large ($> 0.3$), suggesting novel generation rather than memorization.

arises from the fact that non-spatial rules are more implicit and lack explicit cues, such as object positions and lengths, which are readily available in spatial relationships. More experiments for various settings (e.g., other backbone models) are deferred to Appendix E.3, which shows consistent empirical observations that DMs can capture coarse-grained rules but struggle to master fine-grained ones.

**Despite Instabilities, DMs Can Generate Fine-Grained Samples.** While fine-grained rule experiments show DMs generally struggle to exactly satisfy underlying rules, we observe that they can occasionally generate rule-conforming samples in Figure 5(a), albeit with instability. For example, in Task A, there are 10 generated samples that (almost) satisfy the fine-grained rule, i.e., $\frac{l_2 h_1}{l_1 h_2} \in [0.99, 1.01]$. To determine whether these 10 ideal samples originate from DDPM's generation or are merely training data replicas (Somepalli et al., 2023a;b; Wang et al., 2024b), we analyze their memorization behaviors. For Task A, we represent each sample with a 13D vector capturing key features $(l_1, l_2, h_1, h_2)$ and encoding RGB colors of sun, pole, and shadow. We then compute Euclidean distances to their nearest neighbors, considering samples as replicas if the distance is below a given threshold. Figure 5(b) shows rule-conforming generations are not mere duplicates, achieving 100% memorization at a large threshold (0.3). Appendix E.2 shows 10 ideal samples and their nearest neighbors, highlighting differences. This suggests that, although unstable, DMs can generate rule-conforming samples. Inspired by this, Section 5 presents a mitigation strategy with additional guidance to improve generation consistency.

**More Experiments on Real-world Data.** Beyond synthetic tasks, we alsp conduct additional experiments on real-world datasets to further demonstrate that DMs can learn coarse rules but struggle with fine-grained ones. Specifically, we considers the SynMirror (Dhiman et al., 2024)

dataset, which presents objects and their reflections, where rules connect features such as color, size, and shape. We find that DDPM captures coarse rules (e.g., matching colors between objects and reflections) but struggles with fine ones, often producing mismatched shapes. We also construct the Cifar-MNIST dataset, which pairs specific CIFAR and MNIST classes (e.g., Cats/Dogs with 0/1). We observe that DDPM satisfies coarse rules (e.g., consistently generating two digits and two objects), but only 20% of the generations adhere to fine-grained rules that require specific class pairings. These observations on real-world datasets validate the conclusions drawn from synthetic tasks: DM fail to capture fine-grained relationships between features. Detailed experimental results are provided in Appendix E.4.

## 4. DMs' Failure from a Theoretical Perspective

This section provides theoretical explanations for our observed phenomenon - DMs' inability to effectively learn precise rules. Our analysis reveals that without prior knowledge on the hidden rules, DMs trained by minimizing the DDPM loss (Ho et al., 2020) exhibit a constant error in rule conformity, indicating that they cannot accurately learn the ground-truth rule.

We consider the following multi-patch data setup, which has been widely employed for theoretical analysis of classification (Allen-Zhu & Li, 2020; Cao et al., 2022; Zou et al., 2023; Lu et al., 2024), and recently for diffusion models (Han et al., 2024a).

**Definition 4.1** (Data distribution with Inter-Feature Rules)**.** Let $\mathbf{u}, \mathbf{v} \in \mathbb{R}^d$ be two orthogonal feature vectors with unit norm, i.e., $\|\mathbf{u}\| = \|\mathbf{v}\| = 1$ and $\langle \mathbf{u}, \mathbf{v} \rangle = 0$. Let $\zeta$ be a random variable with its distribution $\mathcal{D}_\zeta$ supporting on a bounded domain $[\underline{c}_\zeta, \overline{c}_\zeta]$ for some constants $0 < \underline{c}_\zeta < \overline{c}_\zeta < \infty$. Each image data consists of multiple patches

$$\mathbf{x} = [\mathbf{x}^{(1)\top}, \mathbf{x}^{(2)\top}, ..., \mathbf{x}^{(P)\top}]^\top,$$

$$\text{where} \quad \mathbf{x}^{(1)} = \zeta \mathbf{u}, \ \mathbf{x}^{(2)} = (1 - \zeta)\mathbf{v},$$

and $\mathbf{x}^{(1)}, \mathbf{x}^{(2)}$ are *independent* with the remaining patches.

Definition 4.1 specifies a *inter-feature rule* on the first two patches of the data, requiring that the projection of the first two feature patches along respective feature vectors $\mathbf{u}, \mathbf{v}$ sum up to one, i.e., $\langle \mathbf{x}^{(1)}, \mathbf{u} \rangle + \langle \mathbf{x}^{(2)}, \mathbf{v} \rangle = 1$. Furthermore, we show such a rule will further lead to a structural constraint on the score function. Specifically, let $\mathbf{x}_0 = [\zeta \mathbf{u}^\top, (1 - \zeta)\mathbf{v}^\top, \mathbf{x}^{(3)\top}, ..., \mathbf{x}^{(P)\top}]$ represent an input image. For arbitrary noise scedules $\{\alpha_t, \beta_t\}$, $\mathbf{x}_t = \alpha_t \mathbf{x}_0 + \beta_t \boldsymbol{\epsilon}_t$ represents the noised image at timestep $t$. We derive the score function along the diffusion path as follows.

**Theorem 4.2.** *The score function is* $\nabla \log p_t(\mathbf{x}_t) \ =$

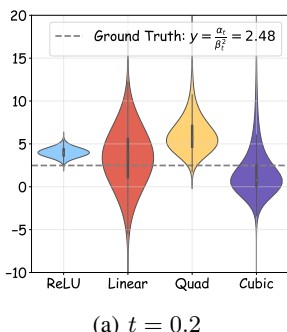
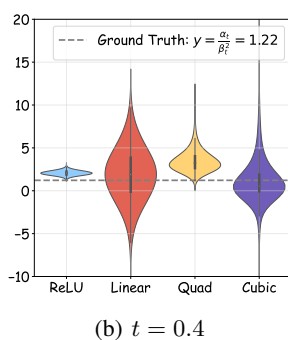
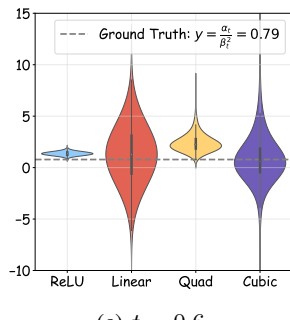
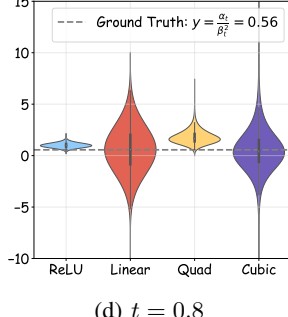

| (a) $t = 0.2$ | (b) $t = 0.4$ | (c) $t = 0.6$ | (d) $t = 0.8$ |

*Figure 6.* **Diffusion model exhibits non-vanishing error on synthetic multi-patch data with norm constraint.** We observe for a variety of timestep $t$ and activation functions (ReLU, linear, quadratic and cubic), a (two-layer) diffusion model cannot learn precisely the hidden norm constraint as in Definition 4.1, with both bias and variance error.

$[\nabla \log p_t(\mathbf{x}_t^{(1)}, \mathbf{x}_t^{(2)})^\top, \nabla \log p_t(\mathbf{x}_t^{(3)}, ..., \mathbf{x}_t^{(P)})^\top]^\top$, *where*

$$\nabla \log p_t(\mathbf{x}_t^{(1)}, \mathbf{x}_t^{(2)})$$
$$= -\frac{1}{\beta_t^2}\mathbf{x}_t + \frac{\alpha_t}{\beta_t^2} \begin{bmatrix} \mathbb{E}_{\mathcal{D}_\zeta}[\pi_t(\zeta, \mathbf{x}_t)\zeta]\mathbf{u} \\ \mathbb{E}_{\mathcal{D}_\zeta}[\pi_t(\zeta, \mathbf{x}_t)(1-\zeta)]\mathbf{v} \end{bmatrix}$$

*where* $\pi_t(\zeta, \mathbf{x}_t) = \frac{\mathcal{N}(\mathbf{x}_t; \boldsymbol{\mu}_t(\zeta), \beta_t^2 \mathbf{I}_{2d})}{\mathbb{E}_{D_\zeta}[\mathcal{N}(\mathbf{x}_t; \boldsymbol{\mu}_t(\zeta), \beta_t^2 \mathbf{I}_{2d})]}$, $\boldsymbol{\mu}_t(\zeta) = [\alpha_t \zeta \mathbf{u}^\top, \alpha_t(1-\zeta)\mathbf{v}^\top]^\top$.

It is clearly noted that the ground truth score (restricted to the first two patches) exhibits the following identity:

$$\mathbb{E}_{\mathcal{D}_\zeta}[\pi_t(\zeta, \mathbf{x}_t)\zeta] + \mathbb{E}_{\mathcal{D}_\zeta}[\pi_t(\zeta, \mathbf{x}_t)(1-\zeta)] = \mathbb{E}_{\mathcal{D}_\zeta}[\pi_t(\zeta, \mathbf{x}_t)]$$
$$= 1. \quad (*)$$

Then, we aim to investigate whether a score network, trained via DSM objective, can accurately conform to such a hidden rule $(*)$. Specifically, we follow (Han et al., 2024a) and consider the following two-layer neural network model: $s_w(\mathbf{x}_t) = [s_w^{(1)}(\mathbf{x}_t)^\top, ..., s_w^{(P)}(\mathbf{x}_t)^\top]^\top$, with

$$s_w^{(p)}(\mathbf{x}_t) = -\frac{1}{\beta_t^2}\mathbf{x}_t^{(p)} + \sum_{r=1}^m \sigma(\langle \mathbf{w}_{r,t}^{(p)}, \mathbf{x}_t^{(p)} \rangle)\mathbf{w}_{r,t}^{(p)}, \quad (1)$$

where each patch is processed with a separate set of $m$ neurons, and $\sigma(\cdot)$ is an (non-constant) polynomial activation function. Such a network mimics the structure of U-Net (Ronneberger et al., 2015) with shared encoder and decoder weights. The network also contains a residual connection that aligns with the score function (Theorem 4.2). Similar network design has been considered in (Shah et al., 2023; Han et al., 2024a). We train the score network by minimizing the DSM loss (Ho et al., 2020; Shah et al., 2023) with expectation on the diffusion noise and the input:

$$L(\mathbf{W}_t) = \mathbb{E}_{\boldsymbol{\epsilon}_t, \mathbf{x}_0} \sum_{p=1}^P \left\| s_w^{(p)}(\mathbf{x}_t^{(p)}) + \frac{\boldsymbol{\epsilon}_t^{(p)}}{\beta_t} \right\|^2 \quad (2)$$

where $\mathbf{x}_t^{(p)} = \alpha_t \mathbf{x}_0^{(p)} + \beta_t \boldsymbol{\epsilon}_t^{(p)}$. It has been shown in (Chen et al., 2022) that minimizing (2) is equivalent to minimizing the score matching loss, i.e., $\mathbb{E}_{p_t} \| s_w - \nabla \log p_t \|^2$.

We next define the *rule-conforming error* to measure the learning outcome of the hidden rule $(*)$.

**Definition 4.3** (Rule-conforming error). For the score network $s_w$ of a diffusion model with weights $\mathbf{w}_{r,t}^{(p)*}$, let

$$\psi_t(\mathbf{x}_t) := \left\langle s_w^{(1)}(\mathbf{x}_t) + \frac{1}{\beta_t^2}\mathbf{x}_t^{(1)}, \mathbf{u} \right\rangle + \left\langle s_w^{(2)}(\mathbf{x}_t) + \frac{1}{\beta_t^2}\mathbf{x}_t^{(2)}, \mathbf{v} \right\rangle$$

be the coefficient along directions $\mathbf{u}, \mathbf{v}$ at time $t$ for $\mathbf{x}_t$. We say the diffusion model conforms to rule $(*)$ if $\psi_t(\mathbf{x}_t) = \frac{\alpha_t}{\beta_t^2}$ holds for *any* $\mathbf{x}_t$. We define the *rule-conforming error* as:

$$\mathcal{E} = \mathbb{E}_{\mathbf{x}_t} \left[ \left( \psi_t(\mathbf{x}_t) - \frac{\alpha_t}{\beta_t^2} \right)^2 \right].$$

Then, we consider training $s_w$ by gradient descent over (2) starting from initialization $\{\mathbf{w}_{r,t}^{(p),0}\}_{r \in [m], p \in [P]}$. The following theorem derives a lower bound on the rule-conforming error for the trained score network model.

**Theorem 4.4.** *Let* $\mathbf{w}_{r,t}^{(p)*}$, $r \in [m]$ *be a stationary point of the DDPM loss* (2). *Then we can lower bound*

$$\mathcal{E} \geq \mathbb{E}_{\zeta, \boldsymbol{\epsilon}_{t,-}^{(1)}} \left[ \text{Var}_{|\zeta, \boldsymbol{\epsilon}_{t,-}^{(1)}} \left( \widetilde{\sigma}^{(1)}(\langle \mathbf{u}, \boldsymbol{\epsilon}_{t,\perp}^{(1)} \rangle) \right) \right]$$
$$+ \mathbb{E}_{\zeta, \boldsymbol{\epsilon}_{t,-}^{(2)}} \left[ \text{Var}_{|\zeta, \boldsymbol{\epsilon}_{t,-}^{(2)}} \left( \widetilde{\sigma}^{(2)}(\langle \mathbf{v}, \boldsymbol{\epsilon}_{t,\perp}^{(2)} \rangle) \right) \right]$$

*where we decompose* $\boldsymbol{\epsilon}_t^{(p)} = \boldsymbol{\epsilon}_{t,-}^{(p)} + \boldsymbol{\epsilon}_{t,\perp}^{(p)}$ *with* $\boldsymbol{\epsilon}_{t,-}^{(p)}$ *being the projection of* $\boldsymbol{\epsilon}_t^{(p)}$ *onto* $\text{span}(\mathbf{w}_{1,t}^{(p),0}, ..., \mathbf{w}_{m,t}^{(p),0})$. $\text{Var}_{(|A)}(\cdot) := \text{Var}(\cdot | A)$ *is the conditional variance and* $\widetilde{\sigma}^{(p)}(\cdot)$ *is a polynomial with coefficients depending on* $\langle \mathbf{w}_{r,t}^{(1)*}, \mathbf{u} \rangle, \langle \mathbf{w}_{r,t}^{(2)*}, \mathbf{v} \rangle$.

Theorem 4.4 immediately suggests a non-vanishing rule-conforming error, as long as the polynomial $\widetilde{\sigma}$ is non-constant and dimension $d$ is sufficiently larger than network width $m$ to ensure variability in the random noise $\boldsymbol{\epsilon}_{t,\perp}$, which is independent of $\mathbf{u}$ and $\mathbf{v}$.

*Remark* 4.5 (Extension to network with patch dependence). Although our neural network setup is separable in the patches, the analysis can be extended to network handling dependent patches. For example, consider $s_w^{(1,2)}(\mathbf{x}_t) = -\mathbf{x}_t^{(1,2)}/\beta_t^2 + \mathbf{W}\sigma(\mathbf{W}^\top \mathbf{x}_t^{(1,2)})$, where $\mathbf{x}_t^{(1,2)}$ denotes the concatenation of two patches, $\sigma(\cdot)$ is a polynomial activation function and $\mathbf{W} \in \mathbb{R}^{2d \times m}$. We can still follow Theorem 4.4 to show that (1) the learned $\mathbf{W}$ consists only the directions $\mathbf{u}, \mathbf{v}$ and its initialization, and (2) the network output becomes a polynomial function of $\langle \mathbf{u}, \boldsymbol{\epsilon}_t^{(1)} \rangle, \langle \mathbf{v}, \boldsymbol{\epsilon}_t^{(2)} \rangle$ and their cross terms (which do not appear in the patch-separated case). As $\mathbf{x}_t$ varies, the function output changes accordingly, which results in a non-vanishing rule conforming error that depends on the variation of $\boldsymbol{\epsilon}_t$.

We now show that when simplifying the model to linear activation $\sigma(x) = x$ and single neuron ($\mathbf{w}_t^{(p)}$), the rule-conforming error can be computed as the sum of bias and variance errors, both of them are lower bounded by some constants. Specifically, we decompose

$$\mathcal{E} = \underbrace{\left| \mathbb{E}_{\mathbf{x}_t}\left[ \psi_t(\mathbf{x}_t) \right] - \frac{\alpha_t}{\beta_t^2} \right|^2}_{\mathcal{E}_{\text{bias}}^2} + \underbrace{\text{Var}\left[ \psi_t(\mathbf{x}_t) \right]}_{\mathcal{E}_{\text{variance}}}.$$

The following theorem suggests there exist a constant bias and variance error for any stationary point $\mathbf{w}_t^*$.

**Theorem 4.6.** *Suppose $\sigma(x) = x$, $m = 1$ and consider $t$ such that $\alpha_t, \beta_t = \Theta(1)$. We train the network with the gradient descent on DDPM loss (2) from small Gaussian initialization, i.e., $\mathbf{w}_t^{(p),0} \sim \mathcal{N}(0, \sigma_0^2 \mathbf{I}_d)$, $\sigma_0 = O(d^{-1/2})$ and $d = \widetilde{\Omega}(1)$. Let $\mathbf{w}_t^{(p)*}$ be any stationary point. Then*

- $\langle \mathbf{w}_t^{(1)*}, \mathbf{u} \rangle, \langle \mathbf{w}_t^{(2)*}, \mathbf{v} \rangle = \Theta(1)$.
- *There exists constants $C_0, C_1 > 0$ (depending on $\mathbb{E}[\zeta], \mathbb{E}[\zeta^2], \alpha_t, \beta_t$) such that $\mathcal{E}_{\text{bias}} = C_0, \mathcal{E}_{\text{variance}} = C_1$.*

Theorem 4.6 shows that (1) all data features $\mathbf{u}$ and $\mathbf{v}$ can be discovered, which is consistent with the results in Han et al. (2024a) and verifies the ability of DMs to conform to coarse rules in the data, i.e., the existence of the key features. (2) It also verifies that DMs fail to learn the fine-grained hidden rule when no constraint or guidance is imposed over the training of DMs. Both of these two results are consistent with our empirical findings in Section 3.

**Empirical verification.** We further train score networks based on the theoretical setup and evaluate the rule-conforming error in Figure 6, where we consider four different activation functions (see Appendix G for details). We calculate the error of DMs in learning the hidden rule (∗) and plot the distribution of $\psi_t(\mathbf{x}_t)$ over 5000 sampled $\mathbf{x}_t$. It is clear that for all activation functions, the rule-conforming error is significant, verifying our theoretical results and suggesting the inability of DMs to precisely learn the hidden

rules.

## 5. Mitigation Strategy with Guided Diffusion

Motivated by our finding that DMs can produce rule-conforming samples but instability, we mitigate this by a common technique, **Guided DDPM**, which introduces additional classifier guidance (Dhariwal & Nichol, 2021) during sampling. Specifically, we train the classifier $f_\theta(\mathbf{x}, t)$ through contrastive learning with constructed contrasting data pairs, where positive samples follow fine-grained rules while negative samples violate fine rules while maintaining coarse-grained compliance. The training objective is

$$\mathcal{L}_{\text{total}} = \mathcal{L}_{\text{classification}} + \lambda \cdot \mathcal{L}_{\text{contrastive}}, \qquad (3)$$

where $\lambda$ is weight parameter, $\mathcal{L}_{\text{classification}}$ is Cross-Entropy loss and $\mathcal{L}_{\text{contrastive}}$ is NT-Xent loss (Sohn, 2016). More details on NT-Xent loss are in Appendix H.1. Then, following Dhariwal & Nichol (2021), gradients from $f_\theta(\mathbf{x}, t)$ are used to guide sampling toward fine-grained rule compliance.

Additionally, based on constructed contrastive data, we directly train a classifier in raw images to determine whether a generation satisfies fine-grained rules. We filter samples predicted as non-rule-conforming to ensure generation quality. This approach, called **Filtered DDPM**, which directly provides guidance based on the noise-free pixel space, can be seen as the upper bound for guided diffusion strategies.

### 5.1. Experiment Results

**Setup.** We use a U-Net classifier $f_\theta(\mathbf{x}, t)$ with guidance weight $\lambda = 1$. Details of the data construction and training process are provided in Appendix H.1.

**Results.** In addition to $R^2$, inspired by the theorical analysis in Section 4, we introduce Error, a metric capturing how well DMs learn hidden rules from variance and bias. Given the Ground Truth line $y = \beta_1 x$ and the Estimation line $\hat{y} = \hat{\beta}_1 x + \hat{\beta}_0$ in Figure 3 and 4, Error is defined as:

$$\text{Error} := \underbrace{|\hat{\beta}_1 - \beta_1| + |\hat{\beta}_0|}_{\text{Bias Error}} + \underbrace{\sqrt{\text{Var}(\hat{y} - y)}}_{\text{Variance Error}} \qquad (4)$$

We measure the bias error $|\mathbb{E}[y - \hat{y}]|$ with the deviation in the estimated coefficients $\hat{\beta}_1, \hat{\beta}_0$. The variance error in (4) corresponds to the square root of $\mathcal{E}_{\text{variance}}$ in Section 4.

Table 2 presents results, Error and $R^2$, before (DDPM) and after applying classifier guidance (Guided DDPM), along with DDPM filtered by pixel-space classifier (Filtered DDPM). Both Guided DDPM and Filtered DDPM outperform the baseline DDPM across all tasks, showing reduced Error and improved $R^2$, with Filtered DDPM achieving the best performance on most tasks.

*Table 2.* **Comparison** between **DDPM**, **Guided DDPM** (Guidance), and **Filtered DDPM** (Filtering): Additional guidance and filtering improve generation with lower Error and higher $R^2$.

| Task | Error ↓ | | | $R^2$↑ | | |
|---|---|---|---|---|---|---|
| | DDPM | Guidance | Filtering | DDPM | Guidance | Filtering |
| A | 0.25 | 0.21 | 0.17 | 0.85 | 0.90 | 0.90 |
| B | 0.11 | 0.10 | 0.05 | 0.83 | 0.85 | 0.86 |
| C | 0.41 | 0.26 | 0.25 | 0.57 | 0.67 | 0.64 |
| D | 0.46 | 0.43 | 0.39 | 0.79 | 0.84 | 0.85 |

### 5.2. Discussions on the Limitation of Guided Diffusion

While guided and filtered diffusion provides some mitigation for rule learning, we acknowledge that this improvement is limited. The limited improvement stems from the inherent nature of our problem: unlike conventional classification tasks, the fine-grained rules that differentiate our contrastive samples exhibit subtle signals, making effective classifier training particularly challenging. In Appendix H.1, we provide additional experimental evidence that, even on such simple synthetic tasks, the classification accuracy on the test set remains between $60\%$ and $80\%$, supporting the difficulty of precise classification in contrastive data.

Additionally, the effectiveness of this strategy relies on prior knowledge of fine-grained rules. In real-world scenarios, fine-grained rules are often difficult to accurately define and detect, making the construction of contrastive data impossible. We leave the solution to DMs' inability to learn fine-grained rules in real-world scenarios for future work.

## 6. Conclusion and Future Work

This study evaluates DMs from the perspective of inter-feature rule learning, revealing through carefully designed synthetic experiments that DMs can capture coarse rules but struggle with fine-grained ones. Theoretical analysis attributes this limitation to a fundamental inconsistency in DMs' training objective with the goal of rule alignment. We further explore some common techniques, such as guided diffusion, to enhance fine-grained rule learning, but observe limited success. Our in-depth findings underscore the inherent difficulty of capturing subtle fine-grained rules, providing valuable insights for future advancements.

One potential direction to address the limitation of DMs in learning fine-grained rules is to introduce stronger reward signals, such as those from human feedback (Ouyang et al., 2022; Xu et al., 2023) or specialized reward models (Fan et al., 2023; Wu et al., 2024), to better guide DMs during generation. However, unlike typical test-time alignment tasks, such as focusing on changing image style or aligning prompts(Wallace et al., 2023; Yang et al., 2024c;a; Yuan et al., 2024a; Kim et al., 2025), fine-grained rule learning is more challenging due to its subtle signal. How to obtain a powerful reward model that accurately reflects various

rules remains an open question. Additionally, incorporating inter-feature rules into the tokenizer—e.g., through masking strategies (He et al., 2021; Assran et al., 2023; Deng et al., 2025)—may help DMs better capture semantic information related to rules. We leave these explorations to future work.

## Acknowledgment

The authors would like to thank the anonymous reviewers and area chair for their helpful comments. Difan Zou and Yujin Han acknowledge the support from NSFC 62306252, Hong Kong ECS award 27309624, Guangdong NSF 2024A1515012444, and the central fund from HKU.

## Impact Statement

This paper aims to advance the fundamental understanding of machine learning, with a focus on diffusion-based image generation. We aim to enhance the interpretability of these models, shedding light on how they manage inter-feature relationships during generation. By deepening our understanding, we hope to drive community attention toward developing image generative methods that align better with real-world needs, such as adherence to physical laws. Acknowledging the societal impact of image synthesis, including its creative potential and risks like image forgery, we strive to promote principled methods and responsible use through our study of these popular diffusion techniques.

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

## A. Low FID and Worse Inter-Feature Learning: A Gaussian Mixture Case

In this section, we provide a toy example based on the Gaussian Mixture Distribution to explain how low FID and incorrect inter-feature relationships can coexist. This supports the point that even though DMs may perform excellently on classical metrics such as FID, this does not necessarily mean they can perfectly learn the hidden inter-feature rules.

Consider a 2-dimensional population, i.e., the true distribution $p(x, y)$, which is a Gaussian Mixture Model (GMM) with two components as:

$$p(x, y) = \mathcal{F}(\boldsymbol{\mu}_p, \boldsymbol{\Sigma}_p) = \frac{1}{2} \cdot \mathcal{N} \left( \begin{bmatrix} 1 \\ 1 \end{bmatrix}, \begin{bmatrix} 1 & 0 \\ 0 & 1 \end{bmatrix} \right) + \frac{1}{2} \cdot \mathcal{N} \left( \begin{bmatrix} -1 \\ -1 \end{bmatrix}, \begin{bmatrix} 1 & 0 \\ 0 & 1 \end{bmatrix} \right). \tag{5}$$

where we can have

$$\boldsymbol{\mu}_p = \frac{1}{2} \cdot \begin{bmatrix} 1 \\ 1 \end{bmatrix} + \frac{1}{2} \cdot \begin{bmatrix} -1 \\ -1 \end{bmatrix} = \begin{bmatrix} 0 \\ 0 \end{bmatrix}$$

and the covaraince matirx as

$$\begin{aligned}
\boldsymbol{\Sigma}_p &= \sum_{i=1}^{2} w_i \left( \boldsymbol{\Sigma}_i + (\boldsymbol{\mu}_i - \boldsymbol{\mu}_q)(\boldsymbol{\mu}_i - \boldsymbol{\mu}_q)^\top \right) \\
&= 0.5 \cdot \left( \begin{bmatrix} 1 & 0 \\ 0 & 1 \end{bmatrix} + \begin{bmatrix} 1 & 1 \\ 1 & 1 \end{bmatrix} \right) + 0.5 \cdot \left( \begin{bmatrix} 1 & 0 \\ 0 & 1 \end{bmatrix} + \begin{bmatrix} 1 & 1 \\ 1 & 1 \end{bmatrix} \right) = \begin{bmatrix} 2 & 1 \\ 1 & 2 \end{bmatrix}.
\end{aligned}$$

We assume the estimated data distribution learned by DMs is a joint Gaussian distribution:

$$q(\hat{x}, \hat{y}) = \mathcal{N}(\boldsymbol{\mu}_q, \boldsymbol{\Sigma}_q) = \mathcal{N} \left( \begin{bmatrix} 0 \\ 0 \end{bmatrix}, \begin{bmatrix} 2 & 1 \\ 1 & 2 \end{bmatrix} \right). \tag{6}$$

With means and covariance matrices of true distributon $p$ and estimated distribution $q$ are identical, that is $\boldsymbol{\mu}_p = \boldsymbol{\mu}_q = [0, 0]^\top$ and $\boldsymbol{\Sigma}_p = \boldsymbol{\Sigma}_q = \begin{bmatrix} 2 & 1 \\ 1 & 2 \end{bmatrix}$, we easily have the FID between $p(x, y)$ and $p(\hat{x}, \hat{y})$ is computed as:

$$\begin{aligned}
\text{FID} &= \left\| \boldsymbol{\mu}_p - \boldsymbol{\mu}_q \right\|_2^2 + \text{Tr} \left( \boldsymbol{\Sigma}_p + \boldsymbol{\Sigma}_q - 2 \left( \boldsymbol{\Sigma}_p \boldsymbol{\Sigma}_q \right)^{1/2} \right) \\
&= \| 0 - 0 \|_2^2 + \text{Tr} \left( \begin{bmatrix} 2 & 1 \\ 1 & 2 \end{bmatrix} + \begin{bmatrix} 2 & 1 \\ 1 & 2 \end{bmatrix} - 2 \begin{bmatrix} 2 & 1 \\ 1 & 2 \end{bmatrix} \right) \\
&= 0
\end{aligned} \tag{7}$$

Although the FID is small (i.e., 0), the inter-feature relationships between $x$ and $y$ in true and estimated distribution are fundamentally different. In the true distribution, $x$ and $y$ are independent within each Gaussian component but exhibit dependence in the overall distribution due to the mixture of components. In the estimated distribution $q(\hat{x}, \hat{y})$, $\hat{x}$ and $\hat{y}$ are dependent with $\text{Cov}(x, y) = 1$. Therefore, low FID does not imply a correct recovery of the inter-feature rules.

## B. On the Rule Conforming Error with linear settings

We remark a recent work (Wang, 2025) analyzes the case when rule-conforming data lies on low-dimensional linear subspace and demonstrates a linear score network can precisely learn the distribution asymptotically. This can be also reflected in our Theorem 4.4, where the polynomial functions $\widetilde{\sigma}^{(1)}(\cdot), \widetilde{\sigma}^{(2)}(\cdot)$ reduce to constant functions, due to the polynomial degree 1 as we consider linear model. This ultimately leads to a zero lower bound for the rule-conforming error.

Nevertheless, our theoretical argument mainly focuses on the general case, where we consider the settings that the underlying rule is *unknown* to the learner, and the model class does *not align* with the true structure of the

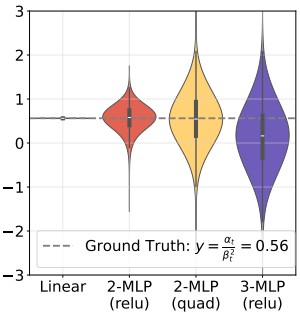

*Figure 8.* Linear network exhibits lower rule-conforming error than more complex networks under the linear rule.

*Table 3.* **Real-World Inter-Feature Rules.** For each scenario containing inter-feature rules, Table 3 provides detailed prompts and annotates the existing inter-feature relationships. By comparing the genuine inter-feature relationships with those in generated images, we can evaluate DMs' ability to learn inter-feature relationships.

---

**[Spatial Rule] (a) Light shadow:**

Prompt 1: `A desert scene with a majestic palace under a bright sun.`

Inter-Feature Rule 1: Sun position affects palace's shadow direction.

Prompt 2: `The moonlight casts a clear shadow of a tall tree onto the ground.`

Inter-Feature Rule 2: Moon position affects tree's shadow direction

---

**[Spatial Rule] (b) Reflection/Refraction**

Prompt 1: `A plush lion toy in front of the mirror. Its front side is facing the camera. There is its reflection in the mirror.`

Inter-Feature Rule 1: The lion toy's orientation relative to the mirror determines its reflection's orientation.

Prompt 2: `A transparent glass bottle partially submerged in a calm, clear pool of water. The upper half of the bottle extends above the water's surface and the lower half of the bottle is submerged.`

Inter-Feature Rule 2: The water surface's position dictates the bottle's shape distortion.

---

**[Spatial Rule] (c) Semantics**

Prompt 1: `A field of sunflowers under a clear blue sky with the sun shining brightly above.`

Inter-Feature Rule 1: Sun direction dictates sunflower orientation.

Prompt 2: `A paintbrush fully loaded with paint, making a stroke on a blank white canvas.`

Inter-Feature Rule 2: Brush tip color matches canvas paint.

---

**[Non-Spatial Rule] (d) Size -Texture**

Prompt 1: `The cross-section of a sturdy tree, covered with annual rings.`

Inter-Feature Rule 1: The diameter of a tree is related to its growth rings.

Prompt 2: `A nautilus fossil, showing its intricate spiral shell structure with visible growth chambers.`

Inter-Feature Rule 2: Nautilus fossil size correlates with spiral patterns.

---

**[Non-Spatial Rule] (e) Size/Region- Color**

Prompt 1: `An artistic representation showing the expanded star phase and cooling star phase of the same star.`

Inter-Feature Rule 1: Celestial body size and color should align, exemplified by red giants and white dwarfs.

Prompt 2: `A burning red candle in a dark with the flame, which is vibrant, dynamic, and glowing intensely against the darkness.`

Inter-Feature Rule 2: Candle flame color varies with distance from the wick.

---

**[Non-Spatial Rule]** (f) Color - Color

Prompt 1: `Two Eclectus parrots, showcasing the striking sexual dimorphism of the species.`

Inter-Feature Rule 1: Eclectus parrots' body and beak colors match—green and yellow for males, red and black for females.

Prompt 2: `A male Poecilia reticulata and a female Poecilia reticulata are swimming gracefully in a clear, freshwater aquarium, showcasing the striking sexual dimorphism of the species`

Inter-Feature Rule 2: Guppies' body and tail colors match—males are equally colorful in both.

---

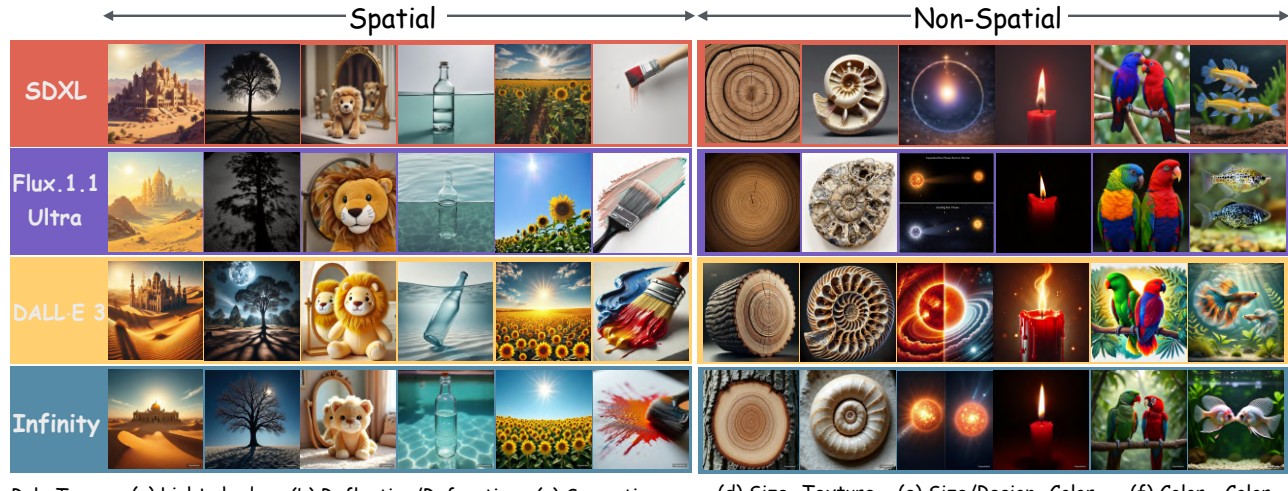

*Figure 7.* **Evaluating More Mainstream DMs on Real-World Inter-Feature Rules.** We evaluate more mainstream DMs on scenarios with inter-feature rules, with 5 random generations and manual selection of unreasonable samples. Despite their success in metrics like FID, none of these DMs achieve complete correctness in cases involving inter-feature relationships.

rule. In such scenarios, although the network may be able to recover the
data distribution, while the hidden rule may still not be well captured. This will still lead to non-zero rule-conforming error.

To further support this claim empirically, we conduct an experiment comparing rule-conforming errors across different model classes in a synthetic linear data setup, where two patches are $\mathbf{x}^{(1)} = \zeta\mathbf{u}$, $\mathbf{x}^{(2)} = -\zeta\mathbf{v}$, with $\zeta \sim \mathcal{N}(0, 1)$. As shown in the Figure 8, linear model achieves significantly lower rule-conforming error compared to more complex, nonlinear models (2-layer, 3-layer MLPs with ReLU or quadratic activation, operated on all patches jointly). This aligns with our findings that without exact structural alignment between the model/objective and the rule, the small rule-conforming error cannot be guaranteed.

## C. Details and More Examples on Real-Wold Hidden Inter-Feature Rules

Table 3 provides a detailed description of the prompts for each case in Figure 1 and Figure 7, including scenarios with inter-feature rules and the corresponding rules themselves. We also consider more DMs such as `SDXL`[4] (Podell et al., 2023), `Flux.1.1 Ultra`[5] (Labs, 2023), `DALL·E 3`[6] (Betker et al., 2023), and VAR-based (Tian et al., 2024a) text-to-image model `Infinity`[7] (Han et al., 2024b) in the evaluation. By comparing these rules, we observe that most mainstream DMs fail in some or all scenarios. For instance, in the *Reflection/Refraction* scenario, none of the DMs successfully generate plausible images: the reflected toy in the mirror faces the camera just like the real one, and the submerged bottle shows no refraction. Our evaluation covers both classic latent diffusion models (e.g., `SD-3.5 Large`) and the latest next-scale prediction-based diffusion models (e.g., `Infinity`). Surprisingly, none of them can perfectly handle these inter-feature relationships, highlighting the widespread limitation of DMs in this regard.

## D. Details and More Example on Synthetic Tasks

This section presents supplementary details and examples regarding our synthetic datasets.

**Task A** generates synthetic images featuring a simple outdoor scene composed of a vertical pole, a sun, and their corresponding shadow. The height of the pole is randomly selected within the range of $[6.4, 12.8]$ pixels, which corresponds to $[20\%, 40\%]$ of the total image size ($32 \times 32$ pixels). The sun's horizontal position is sampled from two predefined distance intervals: far distances ($0 - 6$ pixels or $26 - 32$ pixels) and near distances ($10 - 16$ pixels or $16 - 22$ pixels), ensuring a

---

[4]https://fal.ai/models/fal-ai/fast-lightning-sdxl

[5]https://fal.ai/models/fal-ai/flux-pro/v1.1-ultra

[6]https://chatgpt.com/g/g-iLoR8U3iA-dall-e3

[7]https://github.com/FoundationVision/Infinity?tab=readme-ov-file

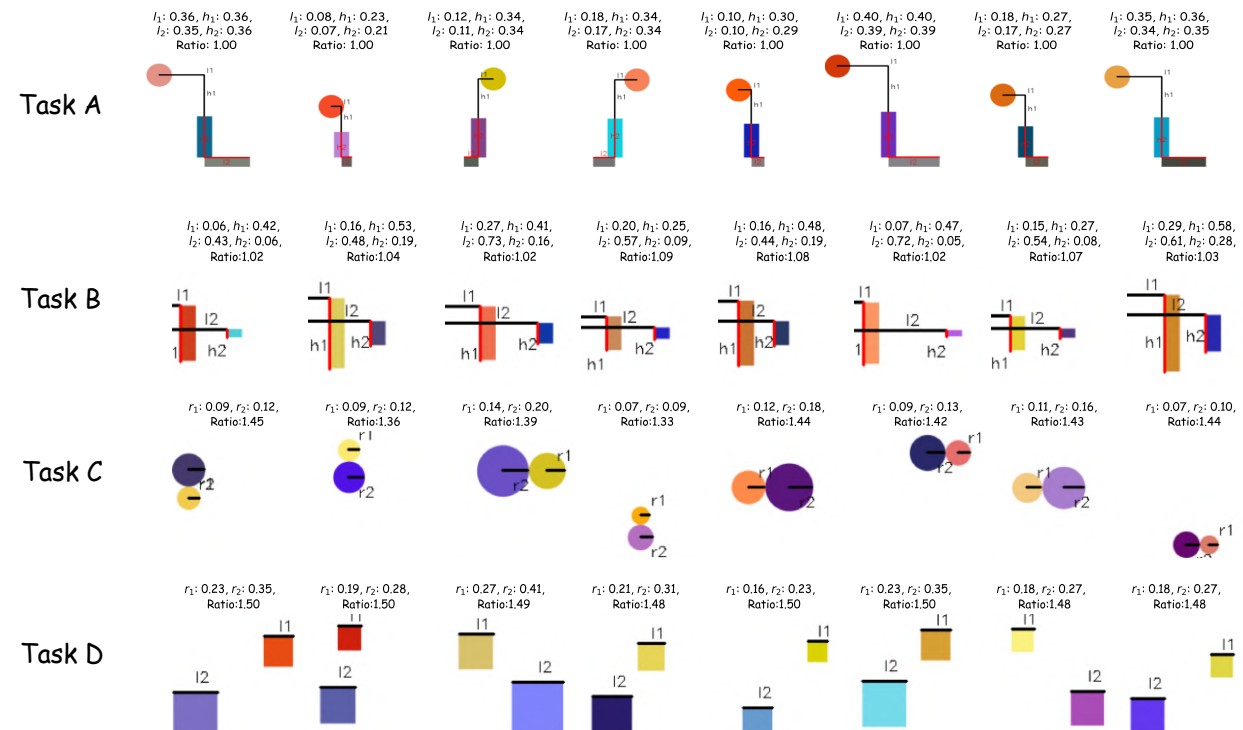

*Figure 9.* **Synthetic Data Examples.** We present synthetic samples in four synthetic tasks, with annotations of features of interest and Ratio calculations. The target Ratios for Tasks A, B, C, and D are 1, 1, $\sqrt{2}$, and 1.5, respectively.

varied distribution of sun locations. The shadow length is computed using the formula:

$$\text{shadow\_length} = \frac{\text{pole\_height} \times |\text{sun\_distance}|}{\text{sun\_height} - \text{pole\_height}} \tag{8}$$

where the sun height is determined as twice the pole height, clipped within $[9.6, 25.6]$ pixels (30%-80% of the image size). Colors for the sun, pole, and shadow are randomly selected from predefined HSV (Hue-Saturation-Value) ranges: Sun color (yellowish tones) has a hue in $[0, 30]$, saturation in $[100, 255]$, and value in $[200, 255]$. Pole color (blue-green tones) has a hue in $[90, 150]$, saturation in $[100, 255]$, and value in $[100, 255]$. Shadow color (dark tones like black, brown, gray) has a hue in $[0, 180]$, saturation in $[0, 50]$, and value in $[50, 150]$.

**Task B** generates synthetic images containing two rectangular objects placed within a $32 \times 32$ pixel space. The first rectangle's position and size are determined as follows: its leftmost position $l_1$ is chosen randomly from the range $[0, 9.6]$ pixels (30% of the image width), and its height $l_2$ is chosen randomly from $[6.4, 19.2]$ pixels (20% to 60% of the image height). The color of the first rectangle is randomly selected from a yellowish hue range with hue $[0, 30]$, saturation $[100, 255]$, and value $[200, 255]$ in HSV space. The second rectangle's position is determined by $h_1$, which is chosen randomly within a range dependent on $l_1$. Specifically, $h_1$ is sampled from the range $[l_1 + 6.4, 25.6]$ pixels (ensuring $h_1 > l_1$). The height of the second rectangle $h_2$ is calculated based on the first rectangle's height $l_2$, ensuring the relation $l_1 h_1 = h_2 l_2$. The color of the second rectangle is chosen randomly from a blue-green hue range with hue $[90, 150]$, saturation $[100, 255]$, and value $[100, 255]$ in HSV space.

**Task C** generates images containing two circles: one large and one small. The large circle's diameter is randomly chosen between 10% and 30% of the image size, and the small circle's diameter is determined to be $\sqrt{2}$ times the diameter of the large circle. The colors of the circles are randomly selected from predefined color ranges in the HSV color space. Specifically, the large circle is assigned a color from the blue-green hue range, with hue values between 90 and 150, saturation between 100 and 255, and brightness between 100 and 255. The small circle is assigned a color from the yellowish hue range, with hue values between 0 and 30, saturation between 100 and 255, and brightness between 200 and 255. The circles are randomly positioned such that they are adjacent to each other—either on the left, right, top, or bottom of the large circle.

**Task D** generates images containing two squares: one smaller and one larger. The small square's size is randomly chosen to

be between 30% and 70% of the top half of the image's size. The larger square's size is then set to be 1.5 times the size of the small square. The color of the small square is randomly selected from a yellowish hue range, with hue values between 0 and 30, saturation between 100 and 255, and brightness between 200 and 255. The color of the large square is randomly chosen from a blue-green hue range, with hue values between 90 and 150, saturation between 100 and 255, and brightness between 100 and 255. The position of the squares is determined within specific regions of the image. The top half and the bottom half of the image are divided into distinct regions, with the small square being placed in the top half and the large square in the bottom half. The exact position of each square is randomly chosen within its respective region, while ensuring that the squares do not exceed the image's boundaries. Both squares are positioned such that they do not overlap with each other and remain entirely within the image frame.

## E. More Synthetic Tasks Setup and Results

### E.1. More Details of Experimental Setup

We use the U-Net architecture as the denoising network, consisting of several down-sampling and up-sampling blocks, each with two convolutional layers followed by ReLU activation. Each down-sampling block incorporates a Self-Attention mechanism and skip connections to preserve fine details. Pooling layers are used to reduce spatial dimensions and capture abstract features. A final $1 \times 1$ convolution layer produces the denoised output image. We use AdamW (Loshchilov, 2017) as the optimizer with a learning rate of $3e - 4$. The noisy steps are set to $T = 1000$, with a linear noise schedule ranging from $1e - 4$ to $2e - 2$. For Tasks A, B, C, and D, the sample sizes are 4000, 2000, 2000, and 2000, respectively, and the input data size is $(3, 32, 32)$. The training is performed on a single NVIDIA A800 GPU for 400, 800, 1600, and 1000 epochs, respectively.

### E.2. More Results of Synthetic Tasks

This section provides additional details to complement the experimental results in Section 3.3. Notably, to ensure more accurate quality assessment of generated images, we upscale the $32 \times 32$ images to $128 \times 128$ during evaluation. This allows the training data to precisely exhibit the expected rule patterns, thereby enabling more reliable evaluation of the generated samples.

**Generations that Violate Coarse-Grained Rules.** Table 1 illustrates the DDPM's ability to learn coarse-grained rules. We observe that in all four synthetic tasks, the number of samples violating the coarse-grained rules is almost zero, except for Task A, where one generated sample, shown in Figure 10, has the sun and shadow on the same side of the pole.

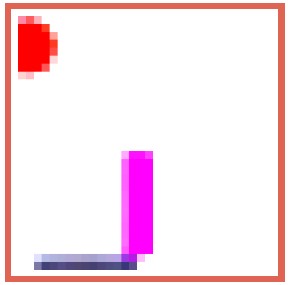

**Generations that Violate Fine-Grained Rules.** We then proceed to show the samples generated by DDPMs that do not satisfy the fine rules in Figure 11, and highlight the features of interest using the evaluation method developed in Section 3.2.

**Generations that Satisfy Fine-Grained Rules.** Here, we use two coordinate systems: a 4D representation capturing key features $(l_1, l_2, h_1, h_2)$ and a 13D representation that additionally encodes the RGB colors of the sun, pole, and shadow. This dual-coordinate analysis allows us to distinguish whether differences between generated and training samples arise from structural variations or merely from different color combinations within similar structures (Okawa et al., 2024). We then compute the Euclidean distances between each generated sample and its nearest neighbor in both 4D and 13D spaces. As a supplement to the DDPM memory experiment in Section 3.3, Figure 12 presents the three nearest neighbors in the training data for high-quality generated samples (with ratios in $[0.99, 1.01]$) in both 4-dimensional and 13-dimensional coordinates. We observe that the 4-dimensional coordinates effectively capture the spatial structure of the nearest neighbors in the training data, while the 13-dimensional coordinates provide a more comprehensive understanding of the similarity of the generated samples, accounting for both color and structure.

*Figure 10.* For Task A, while all training samples have the sun and shadow on opposite sides, DDPM generates one sample violating this coarse-grained rule where the sun and shadow appear on the same side.

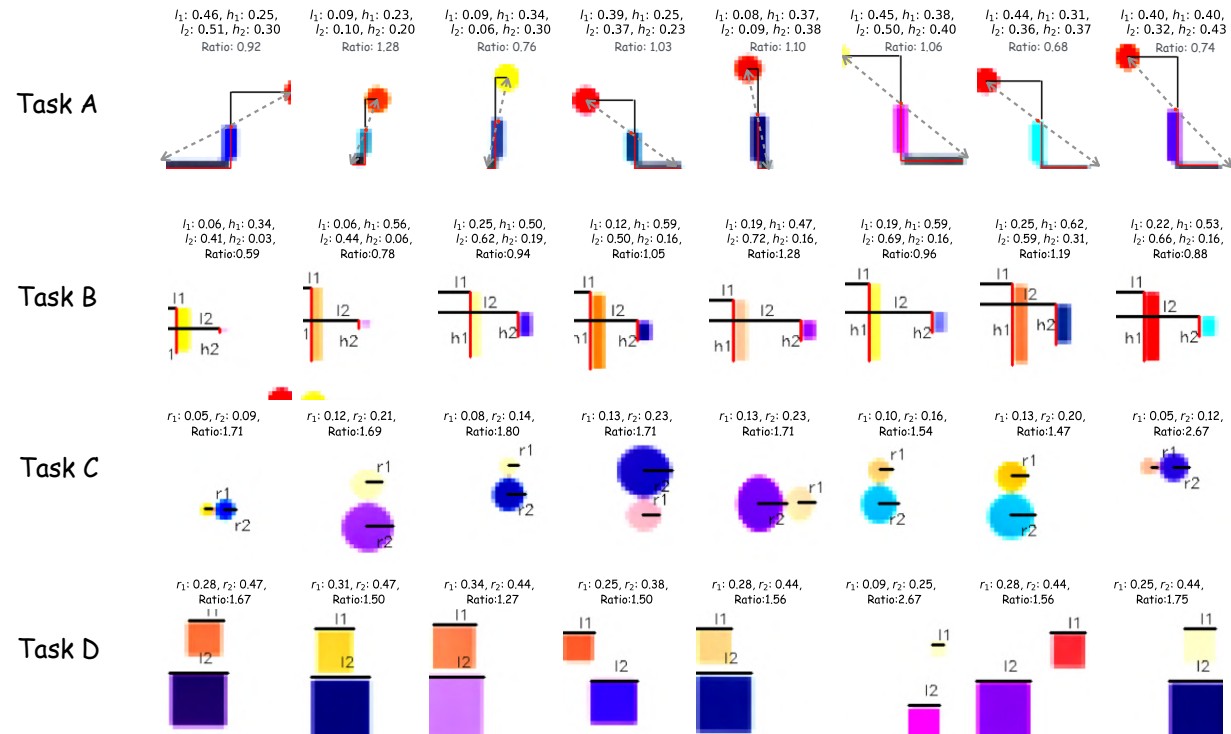

*Figure 11.* **Examples of Rule-violating Generations.** We present samples generated by DDPM that violate fine-grained rules in four synthetic tasks, with annotations of features of interest and Ratio calculations. The target Ratios for Tasks A, B, C, and D are 1, 1, $\sqrt{2}$, and 1.5, respectively.

## E.3. More Settings of Synthetic Tasks

In this section, we consider additional factors, such as more powerful model architectures and larger training datasets, to evaluate the diffusion model's ability to learn precise rules in Task A. Furthermore, detailed experimental results not included in Section 3.3, such as samples that violate coarse rules, will be presented in this section.

**More Training Epochs.** Taking Task A as an example, Figure 13 shows the impact of more training epochs on learning fine-grained rules. We observe that as the number of training epochs increases, the DDPM's ability to learn fine-grained rules improves significantly from 200 to 400 epochs, with $R^2$ increasing from 0.19 to 0.85. This indicates that the relationship between $l_1 h_2$ and $l_2 h_1$ is better described by the linear model. However, even as the training continues up to 4000 epochs, there is no noticeable improvement in the model's ability to learn the fine-grained rules, as reflected by the slight changes in the fitted line coefficients and $R^2$ remaining around 0.85.

**More Model Architectures.** Then, we consider the factor modle architectures and use more powerful backbones, DiT (Peebles & Xie, 2023) and SiT (Ma et al., 2024), to replace U-Net as the denoising network. Specifically, we consider two sizes of DiT and SiT: DiT Small with 33M parameters and patch size (DiT-S/2), DiT Base with 130M parameters and patch size (DiT-B/2), SiT Small with 33M parameters and patch size (SiT-S/2), and SiT Base with 130M parameters and patch size (SiT-B/2). Keeping the number of training epochs, noise time steps, and other hyperparameters consistent, we find that, compared to the 14M parameter U-Net, the parameter count of SiT and DiT has increased by 2 to 10 times. However, as revealed in Figure 14, although all models follow coarse rules, the deficiency in DDPM's ability to learn fine-grained rules does not significantly improve with the increase in parameter count, and there is even a slight decrease in performance with DiT-S/2.

**More Training Data.** Next, we consider the impact of training data size. For Task A, we gradually increase the sample size from 4000 to 20000 to 40000 and observe whether increasing the sample size improves the DMs' ability to learn rules. Figure 15(a) and Figure 15(b) show that the increase in sample size does not enable DMs to learn fine-grained rules better, as evidenced by the almost unchanged $R^2$ and the fitted linear model. Similarly, we do not observe DMs violating coarse

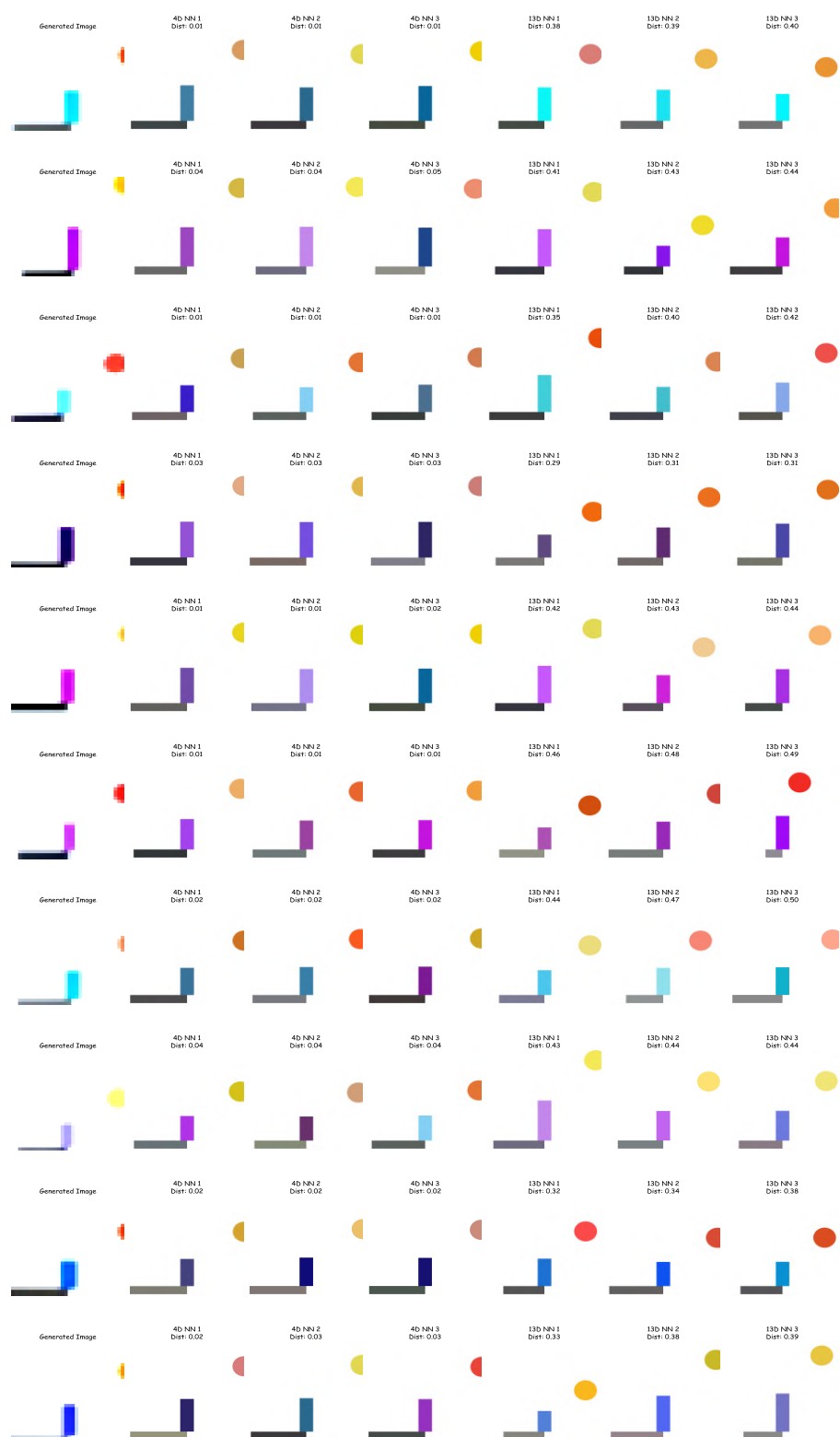

*Figure 12.* **Generations that Violate Fine-Grained Rules.** Taking Task A as an example, we show 10 high-quality generated samples and their Top-3 nearest neighbors from the training data. The first column visualizes the generated samples, while columns 2-4 display the Top-3 nearest neighbors from training data in 4D coordinates, where similarity mainly reflects spatial structure. Columns 5-7 show the top-3 nearest neighbors in 13D coordinates, where similarity primarily reflects object colors.

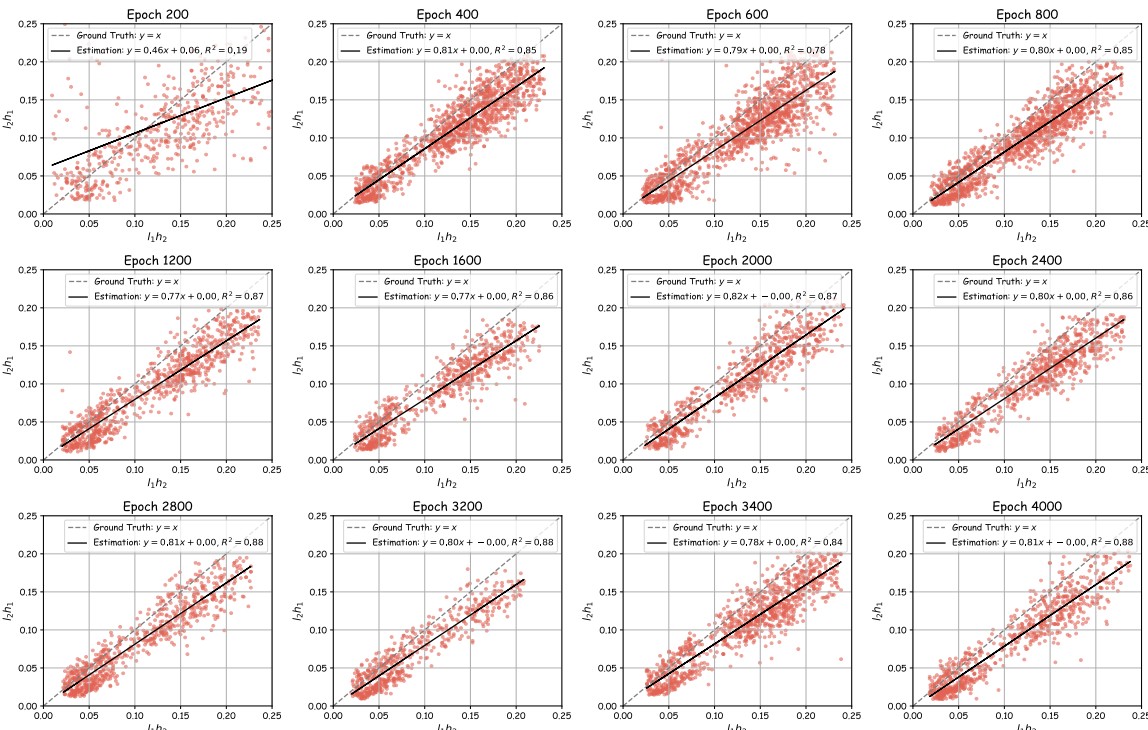

*Figure 13.* **The learning capability of DDPM for fine-grained rules across training epochs.** We observe that as epochs increase from 200 to 4000, DDPM's ability to learn fine-grained rules shows no significant improvement, as evidenced by the stable Estimation line and $R^2$. This suggests that increasing training iterations does not alleviate DMs' difficulty in learning fine-grained rules. The visualized generated samples fall within the interval $[2.5\%, 97.5\%]$.

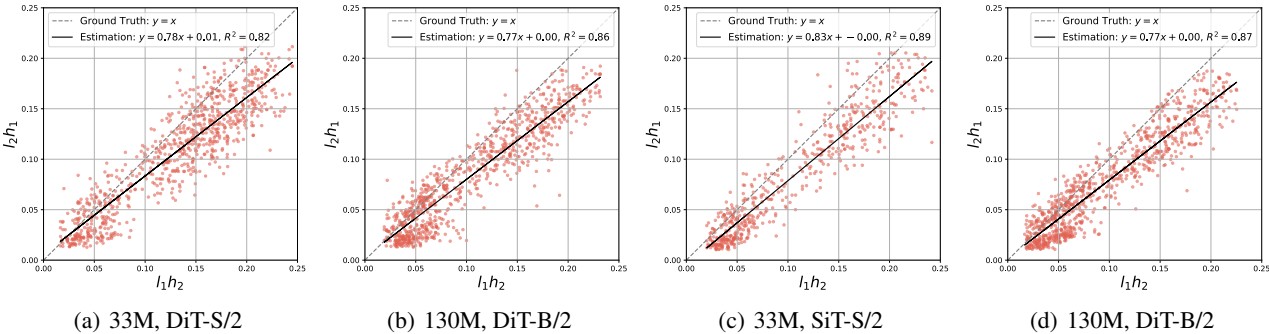

(a) 33M, DiT-S/2      (b) 130M, DiT-B/2      (c) 33M, SiT-S/2      (d) 130M, DiT-B/2

*Figure 14.* **DDPM's capability in learning fine-grained rules with more powerful backbones.** Even with larger and more advanced denoising networks, DDPM still cannot avoid generating samples that violate fine-grained rules. This indicates that DDPM's inability to learn fine-grained rules is decoupled from model architecture. The visualized generated samples fall within the interval $[2.5\%, 97.5\%]$.

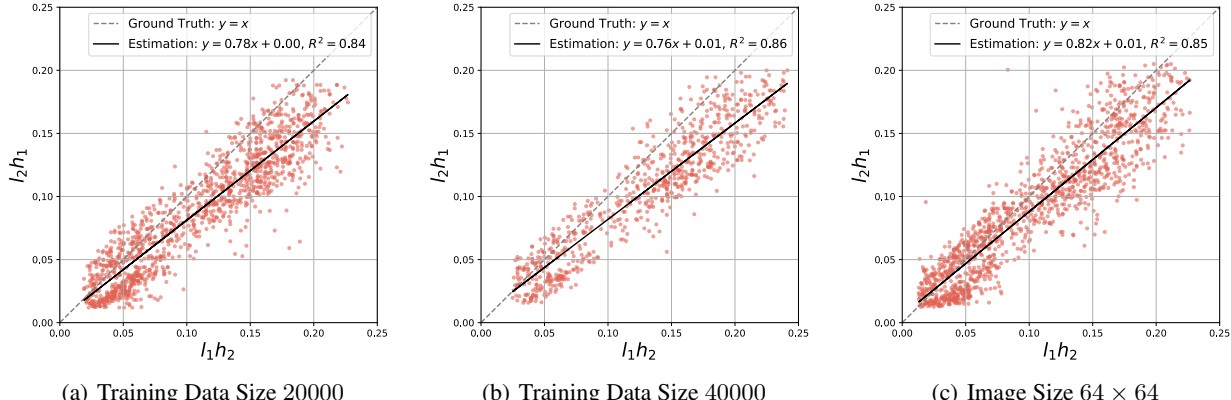

| (a) Training Data Size 20000 | (b) Training Data Size 40000 | (c) Image Size $64 \times 64$ |
|---|---|---|

*Figure 15.* **DDPM's capability in learning fine-grained rules with increased training samples and larger image sizes.** We observe that increasing training samples and image sizes does not significantly improve DDPM's ability to learn fine-grained rules, as evidenced by the stable Estimation line and $R^2$. This suggests that neither expanding the training dataset nor increasing image resolution alleviates DMs' difficulty in learning fine-grained rules. The visualized generated samples fall within the interval $[2.5\%, 97.5\%]$.

rules with large samples.

**More Image Size Choice.** Our final consideration is image size. In the main text, the images are only $32 \times 32$. Existing studies suggest that low-resolution images may lead to the loss of details in diffusion models' generation (Chen et al., 2024; Niu et al., 2024; Li et al., 2024a). Therefore, we consider larger input resolutions of $(3, 64, 64)$, as shown in Figure 15(c). We observe almost no improvement in the DMs' ability to learn underlying rules with generated samples that do not violate coarse rules. Due to computational constraints, we were unable to explore even higher resolutions. But it is clear that for a relatively simple task like Task A, which does not contain rich semantics, DMs are unable to recover the underlying feature relationships even at the $64 \times 64$ resolution. This itself highlights the difficulty DMs face in learning hidden features.

### E.4. More Experiments on Real-world Data

In this section, we train DDPM on real-world datasets to support the observation that diffusion models (DMs) can learn coarse rules but have limited ability to capture fine-grained ones. Specifically, we first consider SynMirror (Dhiman et al., 2024), a mirror dataset where the rules are defined by strict physical relationships between objects and their reflections. Figure 16 shows image generations from DDPM trained on the SynMirror dataset for 400 epochs. We observe that DDPM can learn coarse inter-feature rules—for example, the color and general shape of the objects and their reflections are consistent. However, it fails to capture fine-grained rules, producing errors in the precise position, angle, and detailed contours of the reflections. We further design a synthetic dataset based on CIFAR and MNIST to enable precise quantification. We synthesize 5 new categories by combining specific CIFAR and MNIST classes: (dog, cat, 0, 1), (automobile, truck, 2, 3), (deer, horse, 4, 5), (airplane, bird, 6, 7), and (frog, ship, 8, 9). The inter-feature rule is defined such that only specific categories from CIFAR and MNIST can be combined. We similarly train a DDPM with a U-Net on the CIFAR-MNIST dataset for 1000 epochs. To evaluate whether the generated images comply with the predefined combination rules, we use pretrained classifiers to independently identify the CIFAR and MNIST components within each generated image. As shown in Figure 17, only about 20% of the generated samples satisfy the predefined rules, while over 90% of the training data adhere to them.

### F. Proofs of Section 4

*Proof of Theorem 4.2.* Given the independence, we can write $p_t(\mathbf{x}_t) = p_t(\mathbf{x}_t^{(1)}, \mathbf{x}_t^{(2)})p_t(\mathbf{x}_t^{(3)}, ..., \mathbf{x}_t^{(P)})$. We derive $p_t(\mathbf{x}_t^{(1)}, \mathbf{x}_t^{(2)})$ as follows. First, we notice that $\mathbf{x}_t^{(1)}|\zeta \sim \mathcal{N}(\alpha_t \zeta \mathbf{u}, \beta_t^2 \mathbf{I})$ and $\mathbf{x}_t^{(2)}|\zeta \sim \mathcal{N}(\alpha_t(1-\zeta)\mathbf{v}, \beta_t^2 \mathbf{I})$. Then we obtain

$$p_t(\mathbf{x}_t^{(1)}, \mathbf{x}_t^{(2)}) = \mathbb{E}_{\mathcal{D}_\zeta}[\mathcal{N}(\boldsymbol{\mu}_t(\zeta), \beta_t^2 \mathbf{I}_{2d})]$$

where we denote $\boldsymbol{\mu}_t(\zeta) = [\alpha_t \zeta \mathbf{u}^\top, \alpha_t(1-\zeta)\mathbf{v}^\top]^\top$.

Thus the score can be computed as $\nabla \log p_t(\mathbf{x}_t) = [\nabla \log p_t(\mathbf{x}_t^{(1)}, \mathbf{x}_t^{(2)})^\top, \nabla \log p_t(\mathbf{x}_t^{(3)}, ..., \mathbf{x}_t^{(P)})^\top]^\top$ where

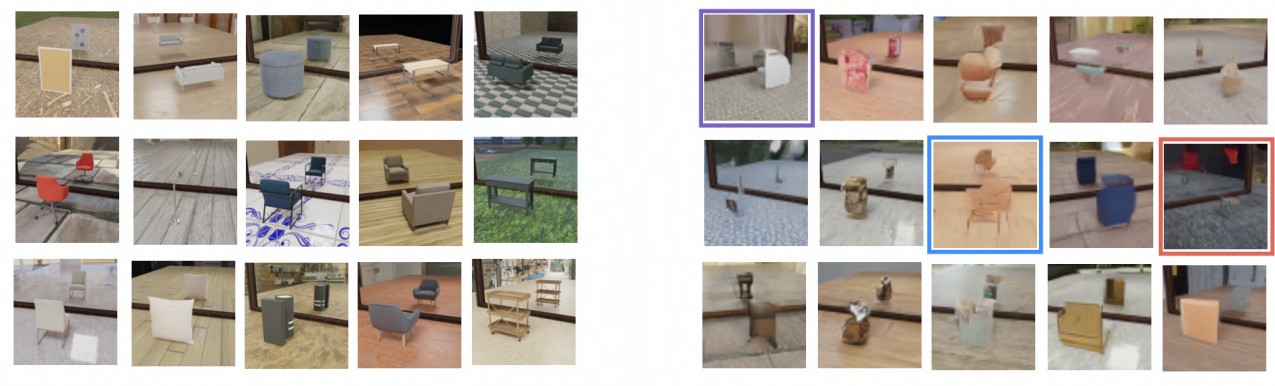

(a) Training SynMirror Data          (b) Generated SynMirror Data

*Figure 16.* **Experimental results on real-world (SynMirror) data. (a)** In SynMirror, the **inter-feature rule** is defined as the object constraining its reflection in the mirror in terms of shape, size, and color. We train a DDPM with a U-Net on SynMirror for 400 epochs. **(b)** The trained DDPM **can capture coarse rules**, such as consistent colors between the object and its reflection, but **struggles with fine rules**, such as unrealistic contours, angles and positions. This highlights DMs' limitation in learning fine-grained rules.

$\nabla \log p_t(\mathbf{x}_t^{(1)}, \mathbf{x}_t^{(2)})^\top \in \mathbb{R}^{2d}$ and can be derived as

$$\nabla \log p_t(\mathbf{x}_t^{(1)}, \mathbf{x}_t^{(2)}) = \frac{\nabla p_t(\mathbf{x}_t^{(1)}, \mathbf{x}_t^{(2)})}{p_t(\mathbf{x}_t^{(1)}, \mathbf{x}_t^{(2)})} = \frac{\mathbb{E}_{\mathcal{D}_\varsigma}[\nabla \mathcal{N}(\mathbf{x}_t; \boldsymbol{\mu}_t(\zeta), \beta_t^2 \mathbf{I}_{2d})]}{\mathbb{E}_{\mathcal{D}_\varsigma}[\mathcal{N}(\mathbf{x}_t; \boldsymbol{\mu}_t(\zeta), \beta_t^2 \mathbf{I}_{2d})]}$$

$$= \frac{\mathbb{E}_{\mathcal{D}_\varsigma}\big[-\mathcal{N}(\mathbf{x}_t; \boldsymbol{\mu}_t(\zeta), \beta_t^2 \mathbf{I}_{2d})\beta_t^{-2}\big(\mathbf{x}_t - \boldsymbol{\mu}_t(\zeta)\big)\big]}{\mathbb{E}_{\mathcal{D}_\varsigma}[\mathcal{N}(\mathbf{x}_t; \boldsymbol{\mu}_t(\zeta), \beta_t^2 \mathbf{I}_{2d})]}$$

$$= -\beta_t^{-2}\mathbf{x}_t + \beta_t^{-2}\mathbb{E}_{\mathcal{D}_\varsigma}[\pi_t(\zeta, \mathbf{x}_t)\boldsymbol{\mu}_t(\zeta)]$$

$$= -\beta_t^{-2}\mathbf{x}_t + \alpha_t \beta_t^{-2} \begin{bmatrix} \mathbb{E}_{\mathcal{D}_\varsigma}[\pi_t(\zeta, \mathbf{x}_t)\zeta]\mathbf{u} \\ \mathbb{E}_{\mathcal{D}_\varsigma}[\pi_t(\zeta, \mathbf{x}_t)(1 - \zeta)]\mathbf{v} \end{bmatrix}$$

where with a slight abuse of notation, we let $\mathbf{x}_t = [\mathbf{x}_t^{(1)\top}, \mathbf{x}_t^{(2)\top}]^\top$ and denote

$$\pi_t(\zeta, \mathbf{x}_t) = \frac{\mathcal{N}(\mathbf{x}_t; \boldsymbol{\mu}_t(\zeta), \boldsymbol{\Sigma}_t)}{\mathbb{E}_{D_\varsigma}[\mathcal{N}(\mathbf{x}_t; \boldsymbol{\mu}_t(\zeta), \boldsymbol{\Sigma}_t)]}.$$

$\square$

*Proof of Theorem 4.4.* According to the decomposition of the rule-respecting error in terms of bias and variance, we have $\mathcal{E}_{\text{mse}} = \mathcal{E}_{\text{bias}}^2 + \mathcal{E}_{\text{variance}}$, where we compute

$$\mathcal{E}_{\text{bias}} = \left| \sum_{r=1}^{m} \mathbb{E}\Big[\sigma\big(\langle \mathbf{w}_r^{(1)}, \mathbf{x}_t^{(1)} \rangle\big)\Big]\langle \mathbf{w}_r^{(1)}, \mathbf{u} \rangle + \sum_{r=1}^{m} \mathbb{E}\Big[\sigma\big(\langle \mathbf{w}_{r,t}^{(2)}, \mathbf{x}_t^{(2)} \rangle\big)\Big]\langle \mathbf{w}_{r,t}^{(2)}, \mathbf{v} \rangle - \frac{\alpha_t}{\beta_t^2} \right|$$

$$\mathcal{E}_{\text{variance}} = \text{Var}\Big( \sum_{r=1}^{m} \sigma(\langle \mathbf{w}_{r,t}^{(1)}, \mathbf{x}_t^{(1)} \rangle)\langle \mathbf{w}_{r,t}^{(1)}, \mathbf{u} \rangle + \sum_{r=1}^{m} \sigma(\langle \mathbf{w}_{r,t}^{(2)}, \mathbf{x}_t^{(2)} \rangle)\langle \mathbf{w}_{r,t}^{(2)}, \mathbf{v} \rangle \Big)$$

where we use the law of total variance and denote $\text{Var}_{|\zeta} = \text{Var}(\cdot|\zeta)$.

Given the gradient direction only consists of $\mathbf{u}, \mathbf{w}_{r,t}^{(1),0}$ and $\mathbf{v}, \mathbf{w}_{r,t}^{(2),0}$ respectively for the two patches, we can decompose the weights $\mathbf{w}_{r,t}^{(1)}, \mathbf{w}_{r,t}^{(2)}$ into

$$\mathbf{w}_{r,t}^{(1)} = \phi_{r,t}\mathbf{w}_{r,t}^0 + \gamma_{r,t}\mathbf{u}$$

$$\mathbf{w}_{r,t}^{(2)} = \varphi_{r,t}\mathbf{w}_{r,t}^0 + \varsigma_{r,t}\mathbf{u}$$

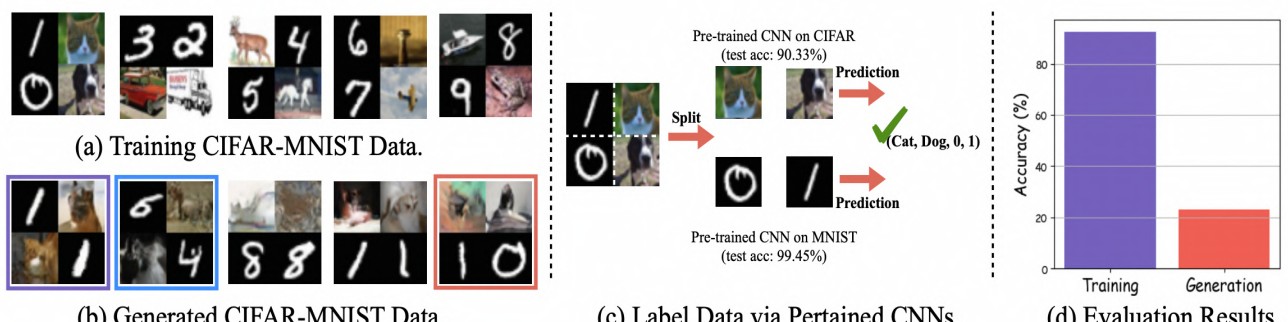

(a) Training CIFAR-MNIST Data.

(b) Generated CIFAR-MNIST Data.

(c) Label Data via Pertained CNNs.

(d) Evaluation Results.

*Figure 17.* **Experimental results on real-world (CIFAR-MNIST) data. (a)** We synthesize 5 new categories by combining specific CIFAR and MNIST classes: (dog, cat, 0, 1), (automobile, truck, 2, 3), (deer, horse, 4, 5), (airplane, bird, 6, 7), and (frog, ship, 8, 9). The **inter-feature rule** is defined such that only specific categories from CIFAR and MNIST can be combined. DDPM with U-Net is trained on CIFAR-MNIST for 1000 epochs. **(b)** The trained DDPM **can capture coarse rules**, such as ensuring each image includes 2 digits and 2 non-digit objects, but **struggles with fine rules**, such as preventing duplicate digits, mismatched digits or mismatched CIFAR-MNIST categories. **(c)** To quantify DDPM's ability to learn fine-grained rules, we use CNNs pre-trained on MNIST and CIFAR to predict labels from 4 sub-images of the generated data. **(d)** Following (c), we evaluate training and generated data by checking if predicted labels fall within the 5 predefined categories. Over 90% of the training data meet this criterion, while only about 20% of the generated data do.

for $r \in [m]$. In addition, we decompose for each $p = 1, 2$

$$\boldsymbol{\epsilon}_t^{(p)} = \boldsymbol{\epsilon}_{t,-}^{(p)} + \boldsymbol{\epsilon}_{t,\perp}^{(p)} = \mathcal{P}_0^{(p)} \boldsymbol{\epsilon}_t^{(p)} + (I_d - \mathcal{P}_0^{(p)}) \boldsymbol{\epsilon}_t^{(p)}$$

where $\mathcal{P}_0^{(p)}$ denotes the projection onto the span of $\{\mathbf{w}_{1,t}^{(p),0}, ..., \mathbf{w}_{m,t}^{(p),0}\}$. Then we can write for the first patch

$$\sum_{r=1}^{m} \sigma(\langle \mathbf{w}_{r,t}^{(1)}, \mathbf{x}_t^{(1)} \rangle) \langle \mathbf{w}_{r,t}^{(1)}, \mathbf{u} \rangle$$

$$= \sum_{r=1}^{m} \sigma(\langle \phi_{r,t} \mathbf{w}_{r,t}^0 + \gamma_{r,t} \mathbf{u}, \alpha_t \zeta \mathbf{u} + \beta_t (\boldsymbol{\epsilon}_{t,-}^{(1)} + \boldsymbol{\epsilon}_{t,\perp}^{(1)}) \rangle) \langle \phi_{r,t} \mathbf{w}_{r,t}^0 + \gamma_{r,t} \mathbf{u}, \mathbf{u} \rangle$$

$$= \sum_{r=1}^{m} \sigma\Big( \phi_{r,t} \alpha_t \zeta \langle \mathbf{w}_{r,t}^0, \mathbf{u} \rangle + \gamma_{r,t} \alpha_t \zeta + \beta_t \langle \phi_{r,t} \mathbf{w}_{r,t}^0 + \gamma_{r,t} \mathbf{u}, \boldsymbol{\epsilon}_{t,-}^{(1)} \rangle + \beta_t \gamma_{r,t} \langle \mathbf{u}, \boldsymbol{\epsilon}_{t,\perp}^{(1)} \rangle \Big) \Big( \phi_{r,t} \langle \mathbf{w}_{r,t}^0, \mathbf{u} \rangle + \gamma_{r,t} \Big)$$

$$= \widetilde{\sigma}^{(1)}(\langle \mathbf{u}, \boldsymbol{\epsilon}_{t,\perp}^{(1)} \rangle)$$

where $\widetilde{\sigma}^{(1)}(\cdot)$ is a polynomial with coefficients depending on $\gamma_{r,t}, \phi_{r,t}, \alpha_t, \beta_t, \zeta$. Similarly, we can write for the second patch that

$$\sum_{r=1}^{m} \sigma(\langle \mathbf{w}_{r,t}^{(2)}, \mathbf{x}_t^{(2)} \rangle) \langle \mathbf{w}_{r,t}^{(2)}, \mathbf{v} \rangle = \widetilde{\sigma}^{(2)}(\langle \mathbf{v}, \boldsymbol{\epsilon}_{t,\perp}^{(2)} \rangle)$$

where $\widetilde{\sigma}^{(2)}(\cdot)$ is a polynomial of the same form as $\widetilde{\sigma}^{(1)}(\cdot)$ except that $\phi_{r,t}, \gamma_{r,t}, \zeta$ is respectively replaced with $\varphi_{r,t}, \varsigma_{r,t}, 1 - \zeta$. Then we can lower bound the variance by

$$\mathcal{E}_{\text{variance}} \geq \mathbb{E}_{\zeta} \Big[ \text{Var}_{|\zeta} \Big( \sum_{r=1}^{m} \sigma(\langle \mathbf{w}_{r,t}^{(1)}, \mathbf{x}_t^{(1)} \rangle) \langle \mathbf{w}_{r,t}^{(1)}, \mathbf{u} \rangle \Big) + \text{Var}_{|\zeta} \Big( \sum_{r=1}^{m} \sigma(\langle \mathbf{w}_{r,t}^{(2)}, \mathbf{x}_t^{(2)} \rangle) \langle \mathbf{w}_{r,t}^{(2)}, \mathbf{v} \rangle \Big) \Big]$$

$$\geq \mathbb{E}_{\zeta, \boldsymbol{\epsilon}_{t,-}^{(1)}, \boldsymbol{\epsilon}_{t,-}^{(2)}} \Big[ \text{Var}_{|\zeta, \boldsymbol{\epsilon}_{t,-}^{(1)}} \big( \widetilde{\sigma}^{(1)}(\langle \mathbf{u}, \boldsymbol{\epsilon}_{t,\perp}^{(1)} \rangle) \big) + \text{Var}_{|\zeta, \boldsymbol{\epsilon}_{t,-}^{(2)}} \big( \widetilde{\sigma}^{(2)}(\langle \mathbf{v}, \boldsymbol{\epsilon}_{t,\perp}^{(2)} \rangle) \big) \Big]$$

where we use law of total variance. $\qquad\qquad\square$

**Lemma F.1** ((Cao et al., 2022)). *If $\mathbf{w}_t^0 \sim \mathcal{N}(0, \sigma_0^2 \mathbf{I})$, we have with probability at least $1 - \delta$*

$$\sigma_0^2 d(1 - \widetilde{O}(d^{-1/2})) \leq \|\mathbf{w}_t^0\|^2 \leq \sigma_0^2 d(1 + \widetilde{O}(d^{-1/2}))$$
$$|\langle \mathbf{w}_t^0, \mathbf{u} \rangle| \leq \sqrt{2 \log(8/\delta)} \sigma_0,$$
$$|\langle \mathbf{w}_t^0, \mathbf{v} \rangle| \leq \sqrt{2 \log(8/\delta)} \sigma_0,$$

*Proof of Theorem 4.6.* Let $L^{(p)}(\mathbf{W}_t) = \mathbb{E}_{\boldsymbol{\epsilon}_t^{(p)}, \mathbf{x}_0^{(p)}} \|s_w^{(p)}(\mathbf{x}_t^{(p)}) + \boldsymbol{\epsilon}_t^{(p)}/\beta_t\|^2$. Then the loss can be written as

$$L(\mathbf{W}_t) = \sum_{p=1}^P L^{(p)}(\mathbf{W}_t) = \sum_{p=1}^P \mathbb{E}_{\boldsymbol{\epsilon}_t^{(p)}, \mathbf{x}_0^{(p)}} \left\| s_w^{(p)}(\mathbf{x}_t^{(p)}) + \frac{1}{\beta_t} \boldsymbol{\epsilon}_t^{(p)} \right\|^2 = \sum_{p=1}^P \mathbb{E}_{\boldsymbol{\epsilon}_t^{(p)}, \mathbf{x}_t^{(p)}} \left\| \langle \mathbf{w}_t^{(p)}, \mathbf{x}_t^{(p)} \rangle \mathbf{w}_t^{(p)} - \frac{1}{\beta_t^2} \mathbf{x}_t^{(p)} + \frac{1}{\beta_t} \boldsymbol{\epsilon}_t^{(p)} \right\|^2$$

We first simplify the loss as follows, where we omit the superscript and consider a single patch due to that each patch is independent and weights are separated.

$$\mathbb{E}_{\boldsymbol{\epsilon}_{t,i}} \left\| \langle \mathbf{w}_t, \mathbf{x}_{t,i} \rangle \mathbf{w}_t - \frac{1}{\beta_t^2} \mathbf{x}_{t,i} + \frac{1}{\beta_t} \boldsymbol{\epsilon}_{t,i} \right\|^2$$

$$= \underbrace{\mathbb{E}_{\boldsymbol{\epsilon}_{t,i}} \left\| \langle \mathbf{w}_t, \mathbf{x}_{t,i} \rangle \mathbf{w}_t \right\|^2}_{I_1} + \underbrace{\mathbb{E}_{\boldsymbol{\epsilon}_{t,i}} \left\| \frac{1}{\beta_t^2} \mathbf{x}_{t,i} - \frac{1}{\beta_t} \boldsymbol{\epsilon}_{t,i} \right\|^2}_{I_2} \underbrace{- 2 \mathbb{E}_{\boldsymbol{\epsilon}_{t,i}} \left[ \langle \mathbf{w}_t, \mathbf{x}_{t,i} \rangle \langle \mathbf{w}_t, \frac{1}{\beta_t^2} \mathbf{x}_{t,i} - \frac{1}{\beta_t} \boldsymbol{\epsilon}_{t,i} \rangle \right]}_{I_3}$$

where we can compute each term following (Han et al., 2024a) as

$$I_1 = \mathbb{E}_{\boldsymbol{\epsilon}_{t,i}}[\langle \mathbf{w}_t, \mathbf{x}_{t,i} \rangle^2] \|\mathbf{w}_t\|^2 = \left( \alpha_t^2 \langle \mathbf{w}_t, \mathbf{x}_{0,i} \rangle^2 + \beta_t^2 \|\mathbf{w}_t\|^2 \right) \|\mathbf{w}_t\|^2, \qquad I_2 = \frac{\alpha_t^2}{\beta_t^4} \|\mathbf{x}_{0,i}\|^2$$

$$I_3 = \frac{\alpha_t}{\beta_t^2} \mathbb{E}_{\boldsymbol{\epsilon}_{t,i}} \left[ \langle \mathbf{w}_t, \mathbf{x}_{t,i} \rangle \right] \langle \mathbf{w}_t, \mathbf{x}_{0,i} \rangle = \frac{\alpha_t^2}{\beta_t^2} \langle \mathbf{w}_t, \mathbf{x}_{0,i} \rangle^2$$

This suggests

$$\mathbb{E}_{\boldsymbol{\epsilon}_{t,i}} \left\| \langle \mathbf{w}_t, \mathbf{x}_{t,i} \rangle \mathbf{w}_t - \frac{1}{\beta_t^2} \mathbf{x}_{t,i} - \boldsymbol{\epsilon}_{t,i} \right\|^2 = \left( \alpha_t^2 \langle \mathbf{w}_t, \mathbf{x}_{0,i} \rangle^2 + \beta_t^2 \|\mathbf{w}_t\|^2 \right) \|\mathbf{w}_t\|^2 - \frac{2\alpha_t^2}{\beta_t^2} \langle \mathbf{w}_t, \mathbf{x}_{0,i} \rangle^2 + I_2$$

where $I_2$ is a constant independent of $\mathbf{w}_t$. Then we obtain the loss for the first two patches as

$$L^{(1)}(\mathbf{w}_t^{(1)}) = \left( \alpha_t^2 \mathbb{E}[\zeta^2] \langle \mathbf{w}_t^{(1)}, \mathbf{u} \rangle^2 + \beta_t^2 \|\mathbf{w}_t^{(1)}\|^2 \right) \|\mathbf{w}_t^{(1)}\|^2 - \frac{2\alpha_t^2}{\beta_t^2} \mathbb{E}[\zeta^2] \langle \mathbf{w}_t^{(1)}, \mathbf{u} \rangle^2 + I_2$$

$$L^{(2)}(\mathbf{w}_t^{(2)}) = \left( \alpha_t^2 \mathbb{E}[(1-\zeta)^2] \langle \mathbf{w}_t^{(2)}, \mathbf{v} \rangle^2 + \beta_t^2 \|\mathbf{w}_t^{(2)}\|^2 \right) \|\mathbf{w}_t^{(2)}\|^2 - \frac{2\alpha_t^2}{\beta_t^2} \mathbb{E}[(1-\zeta)^2] \langle \mathbf{w}_t^{(2)}, \mathbf{v} \rangle^2 + I_2.$$

We next analyze the training dynamics of the gradient descent on the first patch. The second patch follows from similar analysis. For notation clarity, we omit the superscript.

The gradient for the first patch can be computed as

$$\nabla L^{(1)}(\mathbf{w}_t) = \|\mathbf{w}_t\|^2 (2\alpha_t^2 \mathbb{E}[\zeta^2] \langle \mathbf{w}_t, \mathbf{u} \rangle \mathbf{u} + 2\beta_t^2 \mathbf{w}_t) + 2 \left( \alpha_t^2 \mathbb{E}[\zeta^2] \langle \mathbf{w}_t, \mathbf{u} \rangle^2 + \beta_t^2 \|\mathbf{w}_t\|^2 \right) \mathbf{w}_t$$

$$- \frac{2\alpha_t^2}{\beta_t^2} \mathbb{E}[\zeta^2] \langle \mathbf{w}_t, \mathbf{u} \rangle \mathbf{u}.$$

It is noticed that the gradient only consists of directions of $\mathbf{w}_t^0$ and $\mathbf{u}$. It suffices to track the gradient descent dynamics projected to the two directions $\mathbf{u}$ and $\widetilde{\mathbf{w}}_t^0$, where $\widetilde{\mathbf{w}}_t^0 = \mathbf{w}_t^0 - \langle \mathbf{w}_t^0, \mathbf{u} \rangle \mathbf{u}$, i.e.,

$$\langle \mathbf{w}_t^{k+1}, \mathbf{u} \rangle = \langle \mathbf{w}_t^k, \mathbf{u} \rangle - \eta \langle \nabla^{(1)} L(\mathbf{w}_t), \mathbf{u} \rangle$$
$$= \left( 1 + \eta \left( 2\alpha_t^2 \beta_t^{-2} \mathbb{E}[\zeta^2] - 2\alpha_t^2 \mathbb{E}[\zeta^2] \|\mathbf{w}_t^k\|^2 - 4\beta_t^2 \|\mathbf{w}_t^k\|^2 - 2\alpha_t^2 \mathbb{E}[\zeta^2] \langle \mathbf{w}_t^k, \mathbf{u} \rangle^2 \right) \right) \langle \mathbf{w}_t^k, \mathbf{u} \rangle$$
$$\langle \mathbf{w}_t^{k+1}, \widetilde{\mathbf{w}}_t^0 \rangle \|\widetilde{\mathbf{w}}_t^0\|^{-1} = \langle \mathbf{w}_t^k, \widetilde{\mathbf{w}}_t^0 \rangle \|\widetilde{\mathbf{w}}_t^0\|^{-1} - \eta \langle \nabla^{(1)} L(\mathbf{w}_t), \widetilde{\mathbf{w}}_t^0 \rangle \|\widetilde{\mathbf{w}}_t^0\|^{-1}$$
$$= \left( 1 + \eta \left( -4\beta_t^2 \|\mathbf{w}_t^k\|^2 - 2\alpha_t^2 \mathbb{E}[\zeta^2] \langle \mathbf{w}_t^k, \mathbf{u} \rangle^2 \right) \right) \langle \mathbf{w}_t^k, \widetilde{\mathbf{w}}_t^0 \rangle \|\widetilde{\mathbf{w}}_t^0\|^{-1}$$

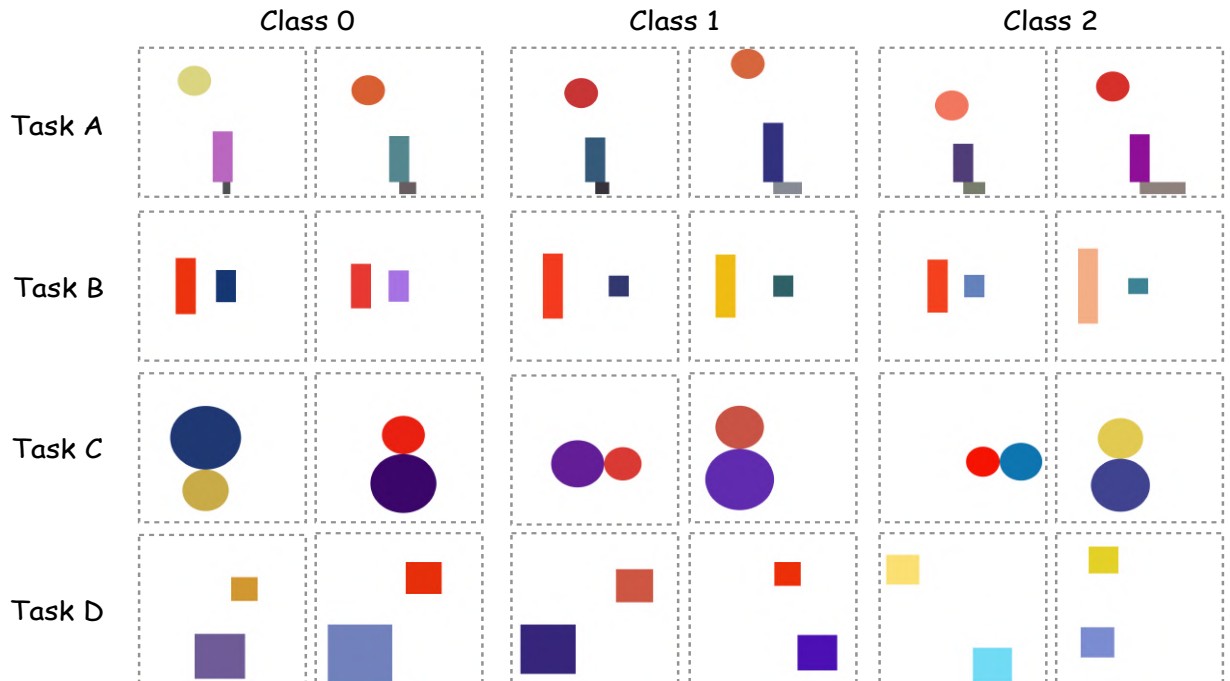

*Figure 18.* **Constructed contrastive data** includes three classes, which differ in fine-grained rules.

It is clear that gradients become zero only when $\Theta(\|\mathbf{w}_t^k\|^2 + \langle\mathbf{w}_t^k, \mathbf{u}\rangle^2) = \Theta(1)$. This suggests that before convergence, $\|\mathbf{w}_t^k\|^2 = o(1)$ given the initialization is small, i.e., $\sigma_0 = O(d^{-1/2})$. We can then verify that $\langle\nabla^{(1)}L(\mathbf{w}_t), \mathbf{u}\rangle \geq \langle\nabla^{(1)}L(\mathbf{w}_t), \widetilde{\mathbf{w}}_t^0\rangle + C$ for some constant $C$.

In addition, suppose we decompose $\mathbf{w}_t^k = \phi_t^k \widetilde{\mathbf{w}}_t^0 + \gamma_t^k \mathbf{u}$, we can see

$$\phi_t^k = \langle\mathbf{w}_t^k, \widetilde{\mathbf{w}}_t^0\rangle\|\mathbf{w}_t^0\|^{-2}, \quad \gamma_t^k = \langle\mathbf{w}_t^k, \mathbf{u}\rangle$$

which then implies

$$\|\mathbf{w}_t^k\|^2 = \langle\mathbf{w}_t^k, \widetilde{\mathbf{w}}_t^0\rangle^2 \|\widetilde{\mathbf{w}}_t^0\|^{-2} + \langle\mathbf{w}_t^k, \mathbf{u}\rangle^2.$$

This combined with the fact that $\langle\nabla^{(1)}L(\mathbf{w}_t), \mathbf{u}\rangle \geq \langle\nabla^{(1)}L(\mathbf{w}_t), \widetilde{\mathbf{w}}_t^0\rangle + C$ suggests that $\langle\mathbf{w}_t^{k+1}, \widetilde{\mathbf{w}}_t^0\rangle\|\widetilde{\mathbf{w}}_t^0\|^{-1}$ cannot increase to $\Theta(1)$ without $\langle\mathbf{w}_t^k, \mathbf{u}\rangle$ reaching $\Theta(1)$. Thus at stationary point, we must have both $\langle\mathbf{w}_t^k, \mathbf{u}\rangle, \|\mathbf{w}_t^k\|^2 = \Theta(1)$.

Next, we analyze the stationary point. Given the gradient only consists of directions $\mathbf{w}_t^k$ and $\mathbf{u}$, we have for any stationary point $\mathbf{w}_t$, it satisfies

$$\langle\nabla L^{(1)}(\mathbf{w}_t), \mathbf{w}_t\rangle = \|\mathbf{w}_t\|^2 (2\alpha_t^2 \mathbb{E}[\zeta^2]\langle\mathbf{w}_t, \mathbf{u}\rangle^2 + 2\beta_t^2\|\mathbf{w}_t\|^2) + 2\left(\alpha_t^2 \mathbb{E}[\zeta^2]\langle\mathbf{w}_t, \mathbf{u}\rangle^2 + \beta_t^2\|\mathbf{w}_t\|^2\right)\|\mathbf{w}_t\|^2$$

$$- \frac{2\alpha_t^2}{\beta_t^2}\mathbb{E}[\zeta^2]\langle\mathbf{w}_t, \mathbf{u}\rangle^2 = 0$$

$$\langle\nabla L^{(1)}(\mathbf{w}_t), \mathbf{u}\rangle = \|\mathbf{w}_t\|^2 (2\alpha_t^2 \mathbb{E}[\zeta^2]\langle\mathbf{w}_t, \mathbf{u}\rangle + 2\beta_t^2\langle\mathbf{w}_t, \mathbf{u}\rangle) + 2\left(\alpha_t^2 \mathbb{E}[\zeta^2]\langle\mathbf{w}_t, \mathbf{u}\rangle^2 + \beta_t^2\|\mathbf{w}_t\|^2\right)\langle\mathbf{w}_t, \mathbf{u}\rangle$$

$$- \frac{2\alpha_t^2}{\beta_t^2}\mathbb{E}[\zeta^2]\langle\mathbf{w}_t, \mathbf{u}\rangle = 0$$

We solve the stationary equalities as

$$\|\mathbf{w}_t^{(1)}\|^2 = 6\alpha_t^2\beta_t^{-2}\mathbb{E}[\zeta^2] + 4 \pm \sqrt{68\alpha_t^4\beta_t^{-4}(\mathbb{E}[\zeta^2])^2 + 80\alpha_t^2\beta_t^{-2}\mathbb{E}[\zeta^2] + 16}$$

$$\langle\mathbf{w}_t^{(1)}, \mathbf{u}\rangle^2 = \frac{1}{\beta_t^2} - \left(\frac{2\beta_t^2}{\alpha_t^2\mathbb{E}[\zeta^2]} + 1\right)\|\mathbf{w}_t^{(1)}\|^2$$

Similarly, we can compute and solve the stationary point for the second patch where $\mathbb{E}[\zeta^2]$ is replaced with $\mathbb{E}[(1-\zeta)^2]$.

We then compute the bias error as

$$
\begin{aligned}
\mathcal{E}_{\text{bias}} &= \mathbb{E}_{\boldsymbol{\epsilon}_t, \mathbf{x}_0}\left[\langle \mathbf{w}_t^{(1)}, \mathbf{x}_t^{(1)}\rangle\langle \mathbf{w}_t^{(1)}, \mathbf{u}\rangle + \langle \mathbf{w}_t^{(2)}, \mathbf{x}_t^{(2)}\rangle\langle \mathbf{w}_t^{(2)}, \mathbf{v}\rangle\right] - \frac{\alpha_t}{\beta_t^2} \\
&= \left|\alpha_t\mathbb{E}[\zeta]\langle \mathbf{w}_t^{(1)}, \mathbf{u}\rangle^2 + \alpha_t\mathbb{E}[1-\zeta]\langle \mathbf{w}_t^{(2)}, \mathbf{v}\rangle^2 - \frac{\alpha_t}{\beta_t^2}\right| \\
&= C_0(\mathbb{E}[\zeta], \mathbb{E}[\zeta^2], \alpha_t, \beta_t)
\end{aligned}
$$

It can be easily verified that there exists a constant bias $C_0$ that depends on $\mathbb{E}[\zeta], \mathbb{E}[\zeta^2], \alpha_t, \beta_t$. In addition, we compute the variance as

$$
\begin{aligned}
\mathcal{E}_{\text{variance}} &= \left(\alpha_t^2\mathbb{E}[\zeta^2]\langle \mathbf{w}_t^{(1)}, \mathbf{u}\rangle^2 + \beta_t^2\|\mathbf{w}_t^{(1)}\|^2\right)\langle \mathbf{w}_t^{(1)}, \mathbf{u}\rangle^2 + \left(\alpha_t^2\mathbb{E}[(1-\zeta)^2]\langle \mathbf{w}_t^{(2)}, \mathbf{v}\rangle^2 + \beta_t^2\|\mathbf{w}_t^{(2)}\|^2\right)\langle \mathbf{w}_t^{(2)}, \mathbf{v}\rangle^2 \\
&\quad + 2\alpha_t^2\mathbb{E}[\zeta(1-\zeta)]\langle \mathbf{w}_t^{(1)}, \mathbf{u}\rangle^2\langle \mathbf{w}_t^{(2)}, \mathbf{v}\rangle^2 - \left(\alpha_t\mathbb{E}[\zeta]\langle \mathbf{w}_t^{(1)}, \mathbf{u}\rangle^2 + \alpha_t\mathbb{E}[1-\zeta]\langle \mathbf{w}_t^{(2)}, \mathbf{v}\rangle^2\right)^2 \\
&= \alpha_t^2\text{Var}(\zeta)\langle \mathbf{w}_t^{(1)}, \mathbf{u}\rangle^4 + \alpha_t^2\text{Var}(1-\zeta)\langle \mathbf{w}_t^{(2)}, \mathbf{v}\rangle^4 - 2\alpha_t^2\text{Cov}(\zeta, 1-\zeta)\langle \mathbf{w}_t^{(1)}, \mathbf{u}\rangle^2\langle \mathbf{w}_t^{(2)}, \mathbf{v}\rangle^2 \\
&\quad + \beta_t^2\|\mathbf{w}_t^{(1)}\|^2\langle \mathbf{w}_t^{(1)}, \mathbf{u}\rangle^2 + \beta_t^2\|\mathbf{w}_t^{(2)}\|^2\langle \mathbf{w}_t^{(2)}, \mathbf{v}\rangle^2 \\
&= \alpha_t^2\text{Var}\left(\zeta\langle \mathbf{w}_t^{(1)}, \mathbf{u}\rangle^2 - (1-\zeta)\langle \mathbf{w}_t^{(2)}, \mathbf{v}\rangle^2\right) + \beta_t^2\|\mathbf{w}_t^{(1)}\|^2\langle \mathbf{w}_t^{(1)}, \mathbf{u}\rangle^2 + \beta_t^2\|\mathbf{w}_t^{(2)}\|^2\langle \mathbf{w}_t^{(2)}, \mathbf{v}\rangle^2 \\
&= C_1(\mathbb{E}[\zeta], \mathbb{E}[\zeta^2], \alpha_t, \beta_t) > 0
\end{aligned}
$$

where we see $\mathbb{E}[A^2] - \mathbb{E}[A]^2 = \text{Var}(A) \geq 0$ for arbitrary random variable $A$. $\qquad\square$

## G. Experiment Details on Synthetic Data with Two-layer Diffusion Model in Section 4

In order to verify the theoretical claims on DMs failing to precisely recover the inter-feature rule ($*$) (in Section 4), we conduct numerical experiments on a two-layer diffusion model on a two-patch data distribution.

Specifically we set $\mathbf{x} = [\mathbf{x}^{(1)\top}, \mathbf{x}^{(2)\top}]$ where $\mathbf{x}^{(1)} = \zeta\mathbf{u}$, $\mathbf{x}^{(2)\top} = (1-\zeta)\mathbf{v}$. Here we set $\mathbf{u} = [1, 0, \cdots 0] \in \mathbb{R}^d$, $\mathbf{v} = [0, 1, 0, \cdots 0] \in \mathbb{R}^d$ with $d = 100$. The score network follows the structure in (1) where we consider $\sigma(\cdot)$ to be ReLU, linear, quadratic and cubic activation functions. We set network width $m = 20$. To simulate the DDPM loss in expectation, for each epoch, we sample $n = 1000$ input data $\mathbf{x}_{0,i}, i \in [n]$ and for each data we sample $n_\epsilon = 1000$ standard Gaussian noise $\boldsymbol{\epsilon}_{t,i,j}, i, j \in [1000]$, and consider minimizing the empirical loss

$$
L(\mathbf{W}_t) = \frac{1}{nn_\epsilon}\sum_{i=1}^{n}\sum_{j=1}^{n_\epsilon}\sum_{p=1}^{2}\|s_w(\mathbf{x}_{t,i,j}^{(p)}) - \boldsymbol{\epsilon}_{t,i,j}^{(p)}\|^2
$$

where $\mathbf{x}_{t,i,j}^{(p)} = \alpha_t\mathbf{x}_{0,i}^{(p)} + \beta_t\boldsymbol{\epsilon}_{t,i,j}^{(p)}, p = 1, 2$. We use gradient descent to train the score network for 5000 epochs. We consider $\alpha_t = \exp(-t)$ and $\beta_t = \sqrt{1 - \exp(-2t)}$ where we set $t = 0.2, 0.4, 0.6, 0.8$. We then check whether learned diffusion models learn the ground-truth rule ($*$) by plotting the distribution of $\psi_t(\mathbf{x}_t)$ against $\alpha_t/\beta_t^2$. The distribution of $\psi_t(\mathbf{x}_t)$ is estimated with 5000 samples $\mathbf{x}_t$.

## H. Details of Mitigation Strategies

### H.1. Details of Guided Diffusion

Guided Diffusion is a common strategy that trains an additional classifier to guide DDPM generation towards desired samples during the sampling process.

**Training Details and Results.** This section includes the details of training classifiers with contrastive learning as guidance. Figure 18 visualizes the constructed contrastive data, where each dataset includes three sample types that differ only in fine-grained rules and appear nearly identical at a glance. Figure 19 visualizes the contrastive datasets constructed for each of the four synthetic tasks. The classifier training for each task is treated as a three-class classification problem with 2000

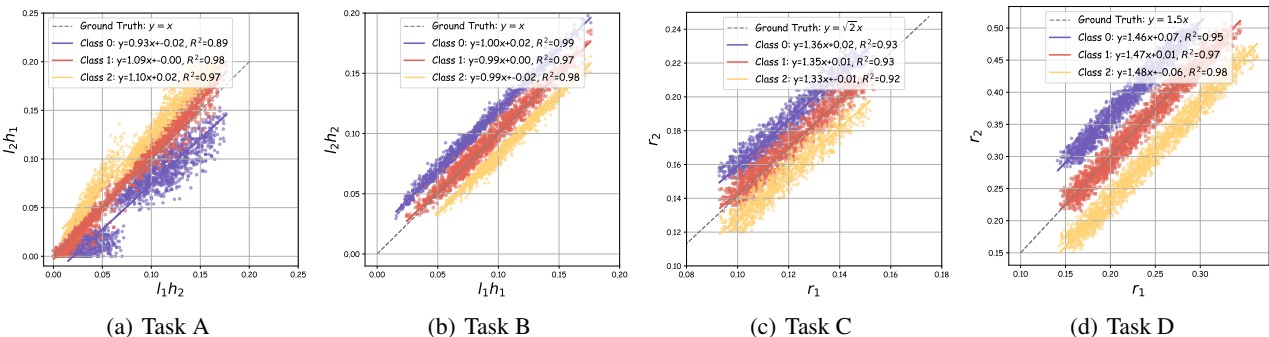

(a) Task A      (b) Task B      (c) Task C      (d) Task D

*Figure 19.* **Construction of contrastive training data.** For each task, we build a three-class dataset where Class 1 represents samples satisfying fine-grained rules, while Classes 0 and 2 represent samples that only satisfy coarse-grained rules. Based on these constructed contrastive datasets, we train classifiers as additional guidance to improve DDPM's generation.

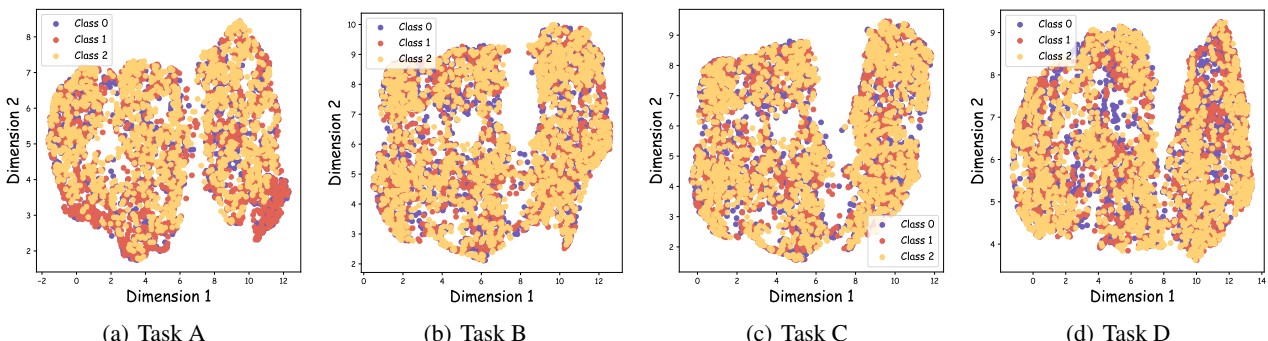

(a) Task A      (b) Task B      (c) Task C      (d) Task D

*Figure 20.* **CLIP representation of contrastive training data**. For each task, we use the CLIP model to extract its representations and apply UMAP for dimensionality reduction. We observe that the contrastive data is nearly inseparable, which presents a challenge for training the classifier.

positive samples (class 1) and 2000 samples per negative class (classes 0 and 2). We use U-Net as the classifier architecture, trained for 20000 iterations with a learning rate of $3e-4$, and a contrastive learning weight $\lambda = 1$. Beyond standard guided diffusion, we dynamically adjust guidance weights (gradient scales) with a piecewise strategy where guidance is activated only in the final 20 denoising steps. The weight linearly increases from 0 to predefined gradient scale factors (7 for Tasks A/C, 10 for Tasks B/D). Through comparison of constant versus piecewise weighting, we report optimal strategies: Task A,B,D for standard sweighting method and Task C employs the piecewise weighting method. As noted in Section 5.2, training high-accuracy classifiers is not easy in our problem, as evidenced by the accuracy of the training data for Tasks A, B, C, and D being 0.57, 0.51, 0.55, and 0.63, respectively.

**NT-Xent Loss.** NT-Xent Loss (Normalized Temperature-scaled Cross Entropy Loss) (Sohn, 2016) is commonly used in contrastive learning to measure the similarity between positive pairs (similar samples) and distinguish them from negative pairs (dissimilar samples).

$$\mathcal{L}_{\text{NT-Xent}}(i, j) = -\log \left( \frac{\exp(\text{sim}(\mathbf{z}_i, \mathbf{z}_j)/\tau)}{\sum_{k=1}^{2N} \mathbf{1}_{[k \neq i]} \exp(\text{sim}(\mathbf{z}_i, \mathbf{z}_k)/\tau)} \right), \tag{9}$$

Where $\mathbf{z}_i$ and $\mathbf{z}_j$ are the embeddings of the $i$-th and $j$-th samples, $\text{sim}(\mathbf{z}_i, \mathbf{z}_j) = \frac{\mathbf{z}_i^\top \mathbf{z}_j}{\|\mathbf{z}_i\|\|\mathbf{z}_j\|}$ is the cosine similarity and $\tau$ is the temperature parameter that scales the similarity which we set $\tau = 0.5$ in our experiments.

### H.2. Details of Filtered DDPM

Filtered DDPM is a more straightforward strategy that uses a classifier trained on raw images to filter DDPM generations, keeping only samples predicted to satisfy fine-grained rules.

**Training Details and Results.** Based on the contrastive data constructed in Figure 19, we split the training and test data in

an 80:20 ratio and directly train a three-way classifier in the raw image space, using MLP, ResNet-8, and U-Net architectures. These models are trained for 100 epochs with a learning rate of $3e - 4$. As shown in Figure 21, the classifiers achieve accuracy between $60\%$ and $80\%$. While they outperform classifiers trained for guided diffusion due to the noise-free setting, they still fail to achieve $100\%$ accuracy, even for these simple synthesis tasks. Additionally, Figure 20 shows the representations extracted by CLIP (Radford et al., 2021) for each synthetic task, followed by dimensionality reduction using UMAP (McInnes et al., 2018). We observe that the data from different categories in the contrastive data is difficult to distinguish, which presents a challenge for training the classifier. Based on test accuracy, we use the trained MLP model to filter DDPM generations for Tasks A and B, keeping only samples predicted as Class 1 (satisfying fine-grained rules). For Tasks C and D, we use the U-Net model for filtering.

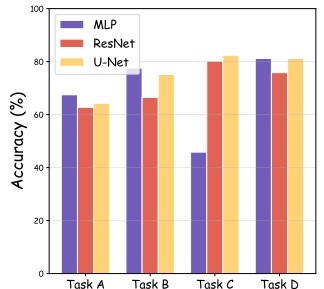

*Figure 21.* Test accuracy of different models on contrastive datasets.

