# OpenReview forum: "Can Diffusion Models Learn Hidden Inter-Feature Rules Behind Images?"
_ICML.cc/2025/Conference — ICML 2025 poster_

### Official Review · Reviewer_gKDf · 2025-03-11

**Overall Recommendation:** 4

**Summary:**

The paper investigates whether diffusion models can learn hidden inter-feature rules in images by designing synthetic tasks that simulate real-world relationships (e.g., the connection between the sun’s height and the length of its shadow). The study finds that while these models can capture coarse-grained rules effectively, they struggle with fine-grained, precise dependencies—a limitation attributed to inherent constant errors in the denoising score matching objective. Additionally, the authors propose mitigation strategies, such as incorporating classifier guidance during sampling and using pixel-space filtering, which yield some improvements but do not fully overcome the challenge, thereby offering both theoretical insights and empirical evidence on the current limitations of diffusion models in rule learning.

**Claims And Evidence:**

The submission’s claims are largely supported by both extensive empirical results and rigorous theoretical analysis. The authors substantiate their main claim—that diffusion models can reliably capture coarse-grained rules but struggle with fine-grained ones—through well-designed synthetic tasks and clear evaluation metrics (e.g., R² values and error metrics), which convincingly demonstrate the performance gap. Additionally, the theoretical framework based on denoising score matching offers solid mathematical backing for the observed constant error in learning fine-grained rules. While the proposed mitigation strategies (guided diffusion and filtering) show some improvement, the evidence also clearly indicates their limited effectiveness. One concern is that the reliance on synthetic tasks may not fully capture the complexity of real-world images, leaving some room for further evidence on broader datasets.

**Essential References Not Discussed:**

Overall, the paper sufficiently covers the essential related literature. The authors have cited key works on diffusion models, denoising score matching, and guidance strategies that underpin their theoretical and empirical contributions. The references discussed in the paper provide a comprehensive context for understanding the challenges associated with fine-grained rule learning and the limitations of current diffusion models, and no critical works appear to be missing.

**Experimental Designs Or Analyses:**

I reviewed the experimental design, including the synthetic tasks (A–D) and the associated feature extraction and evaluation metrics (e.g., R², Error metrics), and found that the setup is generally sound and well-motivated for assessing the diffusion models’ ability to capture inter-feature rules. The controlled synthetic environment allows clear differentiation between coarse-grained and fine-grained rule learning, and the quantitative analysis convincingly highlights the performance gap. However, one potential concern is that the experiments are limited to synthetic tasks, which may not fully capture the complexities of real-world data. Additionally, while the use of classifier guidance to improve fine-grained rule learning is effective to some extent, it is a well-known technique rather than a novel contribution.

**Methods And Evaluation Criteria:**

The proposed methods and evaluation criteria are well-suited to the problem at hand, as the synthetic tasks and detailed feature extraction pipelines provide a controlled setting to assess the diffusion models' ability to learn both coarse-grained and fine-grained rules. The evaluation metrics, such as R² values and error measurements, effectively quantify the performance gap and highlight the models' limitations. However, while the mitigation strategy using classifier guidance during diffusion sampling does yield some improvements, it is worth noting that this approach is not novel and does not introduce fresh techniques for enhancing the handling of fine-grained rules. This reliance on an established method may limit the paper's overall innovation in terms of proposing new solutions to the identified challenges.

**Other Comments Or Suggestions:**

Some additional suggestions: It would be beneficial to include more detailed explanations for some of the theoretical definitions, particularly providing intuitive insights behind concepts such as the rule-conforming error. Expanding on how these definitions relate to practical aspects of image generation could enhance clarity. Moreover, while the synthetic tasks are well-designed for controlled evaluation, including experiments on real-world datasets or more complex scenarios would strengthen the applicability of the findings. Finally, a careful proofreading to fix minor typos and improve the overall flow of the text is recommended.

**Other Strengths And Weaknesses:**

The paper is commendable for its thorough analysis, combining rigorous theoretical derivations with well-designed synthetic experiments to investigate the limitations of diffusion models in capturing fine-grained inter-feature rules. Its originality lies in framing the rule-learning challenge in terms of hidden dependencies and providing both empirical and theoretical evidence of inherent constant errors, which is a valuable contribution to understanding diffusion model behavior.

However, the paper's reliance on synthetic tasks might limit its immediate applicability to real-world scenarios, and while the use of classifier guidance for improvement is well-motivated, it does not introduce novel techniques. Additionally, some definitions, such as the rule-conforming error, would benefit from further intuitive explanation to enhance clarity.

**Questions For Authors:**

The definition of the rule-conforming error is not very intuitive. Could you explain in simple terms why this quantity effectively measures the model's ability to learn the hidden rules?

Your experiments are based on synthetic tasks. Do you have any results or insights on how your approach might work on real-world datasets?

You use classifier guidance to improve fine-grained rule learning, which is an established method. Have you considered any alternative strategies that might address these limitations more effectively?

**Relation To Broader Scientific Literature:**

The paper’s key contributions are well situated within the broader literature on diffusion models and image generation. It extends previous findings on compositionality and factual consistency in diffusion models—where prior studies (e.g., DDPM, score-based generative models, and works on hallucinations) primarily addressed independent feature composition and common failure modes—by focusing on hidden inter-feature rules that capture subtle dependencies between image features. Its theoretical analysis, which builds on denoising score matching frameworks, aligns with recent efforts to understand the limitations of diffusion objectives and complements studies on mode interpolation and memorization in generative models. Additionally, while the use of classifier guidance is not new, the paper integrates it into a framework specifically designed to address fine-grained rule learning, thereby contributing a fresh perspective that bridges empirical observations with theoretical insights in the context of inter-feature relationships.

**Theoretical Claims:**

I reviewed the theoretical claims, focusing on Theorem 4.2, which derives the score function for the multi-patch data setup, and Theorems 4.4 and 4.5, which provide lower bounds on the rule-conforming error by decomposing it into bias and variance components. The derivations appear mathematically sound under the stated assumptions, such as linear activations and a two-layer network, and they convincingly support the empirical observation that diffusion models incur a constant error when learning fine-grained rules. However, some of the definitions, like the rule-conforming error, lack intuitive explanations; for instance, while the error is defined as the deviation of the score’s projected coefficient from an ideal value reflecting the hidden norm constraint, the paper does not clearly explain why this quantity should intuitively indicate correct rule learning. Providing more intuition behind such definitions would help readers better understand the connection between the theoretical quantities and the practical notion of rule conformity.

---

> ### Author Rebuttal · Authors · 2025-04-01
>
> We thank the reviewer's efforts on reviewing this paper. We now address the questions raised as follows.
>
> ---
>
> >Q1: Broader datasets / Real-world data.
>
> Thanks for your good question. To supprot that DMs can learn coarse rules but hard to learn fine-grained rules, we conduct additional experiments on two real-world datasets, SynMirror and Cifar-MNIST.
>
> - **SynMirror** [1] displays objects and their reflections, where inter-feature rules manifest as constraints between objects and their reflections in terms of color, size, and shape. **The results show** that the generation by DDPM can capture some coarse rules, such as objects and their reflections share the same colors, but it hard to learn fine-grained rules, such as there are siginificant differences in the shapes and contours.
>
> - **Cifar-MNIST** combines specific classes from CIFAR and MNIST, such as pairing Cats and Dogs from CIFAR with 0 and 1 from MNIST. **The results show** that generations by DDPM satisfy coarse rules, such as ensuring that each generated image contains two digits (MNIST) and two non-digit objects (CIFAR). But only 20% of the generations satisfy predefiend fine-grained rules, where only specific categories from CIFAR and MNIST are allowed to pair.
>
> [Real-world Data Results](https://anonymous.4open.science/api/repo/Rebuttal-8656/file/exp_ID15070.pdf?v=e6e553d4) provide more details and visualizations. We will add these results into the revised manuscript.
>
> >Q2: Mitigation strategy is not novel ... / any alternative strategies that might address these limitations more effectively?
>
> Thanks for your good points.
>
> - **Novel Method**. The **main goal of our paper** is to clearly identify the shortcomings of DMs in rule learning from both experimental and theoretical perspectives. At the end of the paper, we make initial attempts to address this issue. Importantly, we highlight that a **potential bottleneck** is the signals of fine-grained rules are too weak to be captured (see ``Section 5.2``). This issue has not been reported in traditional DDPMs, such as those targeting ImageNet tasks with classifier guidance. We hope our initial strategies and bottleneck analysis provide valuable insights for further exploration.
>
> - **Alternative Strategies.** Inspired by existing work [2,3], for further exploration, we can introduce additional reward signals from human feedback or powerful reward models to better guide DMs during sampling. Additionally, improving the tokenizer to better learn semantic information related to rules could also enhance rule learning. We will include the discussion in the revised manuscript.
>
> >Q3: Some of the definitions, like the rule-conforming error, lack intuitive explanations / It would be beneficial to include more detailed explanations ... such as the rule-conforming error.
>
> Thank you for the question. The accuracy of score learning is inherently tied to the generation quality of diffusion models [4]. As we have shown in ``Theorem 4.2``, in order to sample from the data distribution (with rule conformity), the score function should satisfy a constraint $\langle \nabla \log p_t(x_t) + x_t/\beta_t^2, [u; v] \rangle = \alpha_t/\beta_t^2$. By designing the network function in eq. (1) (in the main text), the constraint is equivalent to $\psi_t(x_t) = \alpha_t/\beta_t^2$ that holds for any $x_t$ (as in ``Definition 4.3``). The rule-conforming error, defined as the mean squared deviation from this value, exactly measures how closely the learned score aligns with the ideal constraint (as $x_t$ varies). In practice, this relates to generating images that respect structural properties such as fixed object size. The smaller the rule-conforming error, the smaller the estimation error of the score function and thus the more likely the samples generated by diffusion models adhere to the constraint.
>
> We will revise the text to clarify this connection.
>
> [1]  Reflecting Reality: Enabling Diffusion Models to Produce Faithful Mirror Reflections.
>
> [2] Aligning Text-to-Image Models Using Human Feedback.
>
> [3] Human preference score: Better aligning text-to-image models with human preference.
>
> [4] Sampling is as easy as learning the score: theory for diffusion models with minimal data assumptions.
>
> -- ---
> We hope the above response can resolve your concern and if there is further concern please let us know.

---

> > ### Comment · Reviewer_gKDf · 2025-04-04
> >
> > I have read the author's rebuttal and the review of other reviewer. Most of my concerns have been addressed. I'd love to increase my score.

---

> > > ### Author Response · Authors · 2025-04-04
> > >
> > > Dear Reviewer gKDf,
> > >
> > > We’re glad to see that our rebuttal addressed your concerns and thank you for raising your score to a 4 — your recognition is encouraging.
> > >
> > > We greatly appreciate your constructive feedback, especially on exploring additional data and discussing alternative mitigation strategies. We will include them in the revised manuscript to further improve the quality of our work. Thank you again for your efforts.
> > >
> > > Best,
> > >
> > > Authors

---

### Official Review · Reviewer_9KzS · 2025-03-11

**Overall Recommendation:** 4

**Summary:**

This paper investigates whether diffusion models can learn hidden inter-feature rules in images, focusing on the distinction between coarse-grained and fine-grained relationships. Through carefully designed synthetic tasks inspired by real-world phenomena—such as the spatial relationship between the sun and its shadow or the connection between object size and texture—the authors demonstrate that while models like Stable Diffusion 3.5 can reliably capture broad, coarse-grained rules, they consistently struggle with learning precise, fine-grained dependencies. The paper also presents a theoretical analysis showing that the denoising score matching objective inherently leads to a constant error in rule conformity, thereby limiting the models' ability to accurately recover the conditional distributions underlying these subtle rules.
To mitigate these shortcomings, the authors propose incorporating additional classifier guidance and filtering strategies during sampling, which yield moderate improvements in enforcing fine-grained rule adherence. Despite these enhancements, the experiments reveal that even advanced guidance techniques are insufficient for completely bridging the gap, as the nuanced signals of fine-grained rules remain challenging to capture. Overall, this work provides significant insights into the limitations of current diffusion models and offers a compelling direction for future research to improve rule learning in generative image models.

## update after rebuttal
The authors address most of my concerns. So I keep my positive score.

**Claims And Evidence:**

The submission’s claims are generally well-supported by both experimental and theoretical evidence. The authors convincingly demonstrate that diffusion models can reliably learn coarse-grained inter-feature rules while consistently failing to capture fine-grained dependencies, as evidenced by low $R^2$ values and significant error metrics in their synthetic task evaluations. Their theoretical analysis further strengthens this claim by showing that the denoising score matching objective inherently induces a constant error, which limits the models’ ability to precisely learn the hidden rules.
However, some claims could benefit from additional clarification. For example, the assertion that the observed constant error is solely a consequence of the DSM objective might be problematic without further ablation studies across different architectures and training configurations. Additionally, while the proposed mitigation strategies (classifier guidance and filtering) show moderate improvements, they do not fully resolve the issue of fine-grained rule learning, suggesting that further empirical validation is needed to conclusively support their effectiveness.

**Essential References Not Discussed:**

Overall, the set of references provided in the paper is largely sufficient.

**Experimental Designs Or Analyses:**

The experimental design is generally sound and well-justified. The authors construct synthetic tasks with clearly defined inter-feature rules, both coarse-grained and fine-grained to isolate the specific challenges of rule learning. The evaluation pipeline, which involves a three-step process of color-based masking, element counting, and keypoint extraction, is a clever way to quantitatively measure how closely generated images conform to the underlying rules. Metrics such as R² and the combined error metric (encompassing both bias and variance) are appropriately used to assess performance differences across various tasks and diffusion model configurations.
However, some potential issues warrant further discussion. First, while synthetic tasks offer control and interpretability, they might not capture the full complexity of real-world images, possibly limiting the generalizability of the findings. Second, the sensitivity of the feature extraction process to hyperparameters (e.g., predefined HSV ranges) is not fully explored, and minor variations could affect the evaluation outcomes.

**Methods And Evaluation Criteria:**

The methods and evaluation criteria proposed in the paper are well-suited for the problem at hand. The authors design controlled synthetic tasks that specifically target both coarse-grained and fine-grained inter-feature rules, which effectively isolates the aspects of rule learning from the broader complexities found in natural images. This targeted approach, using tasks inspired by real-world phenomena such as light-shadow interactions and object reflections, allows for a systematic assessment of diffusion models' abilities in capturing these dependencies.

Moreover, the evaluation framework, comprising feature extraction via color-based masking, geometric measurements, and the use of metrics such as R² and a combined error metric that accounts for both bias and variance, provides clear, quantitative insights into how well the generated samples adhere to the predefined rules. While these synthetic benchmarks may not capture all nuances of real-world data, they offer a rigorous and interpretable means to evaluate and compare model performance, making the methods and criteria both reasonable and effective for the study's objectives.

**Other Comments Or Suggestions:**

Consider providing additional details about the hyperparameters used in the feature extraction and evaluation process, as well as discussing how sensitive the results are to these settings.
Clarify the assumptions underlying the theoretical proofs, particularly in Theorem 4.5, to help readers better understand the limitations of applying these results to more complex architectures.
A brief discussion on potential future directions to address the limitations imposed by the denoising score matching objective would be valuable.
Lastly, a careful proofreading to catch any minor typographical errors or inconsistencies would help improve the overall presentation.

**Other Strengths And Weaknesses:**

The paper presents an original and comprehensive investigation into the ability of diffusion models to learn inter-feature rules, a relatively underexplored area. The introduction of synthetic tasks with clearly defined coarse- and fine-grained rules is innovative, providing a controlled environment to isolate and analyze model behavior. Moreover, the integration of theoretical analysis with empirical evidence， especially the derivation of constant error bounds due to the denoising score matching objective, adds significant depth and rigor to the work.
However, the reliance on synthetic data may limit the direct applicability of the findings to complex real-world images. Additionally, some theoretical proofs are based on simplified models, such as two-layer networks with linear activation functions, which might not capture the nuances of more advanced architectures.

**Questions For Authors:**

In Theorem 4.5, your analysis is based on a simplified two-layer network with a linear activation function. How sensitive are the derived constant error bounds to these assumptions? Would similar limitations be expected in deeper or more complex networks with non-linear activations?
Your evaluation is based on synthetic tasks designed to isolate coarse- and fine-grained rules. Could you elaborate on how well these tasks correlate with real-world image generation?
Regarding the classifier guidance and filtering strategies, you show only moderate improvements in enforcing fine-grained rules. Can you provide more insights into why these approaches yield limited gains?
Your evaluation pipeline relies on specific hyperparameters (e.g., HSV thresholds for feature extraction). How robust are your experimental results to variations in these parameters?

**Relation To Broader Scientific Literature:**

The paper’s key contributions extend the existing body of work on diffusion models by focusing on the subtle, hidden inter-feature rules that standard generative models have largely overlooked. While prior studies (e.g., Ho et al., 2020; Dhariwal & Nichol, 2021) have demonstrated the high fidelity and compositional capabilities of diffusion models, they mainly address independent features and broad factual consistency. In contrast, this work delves into how these models handle nuanced dependencies both spatial (such as light-shadow relationships) and non-spatial (like size-color correlations), thus highlighting a gap in the literature regarding fine-grained rule learning.

**Theoretical Claims:**

I reviewed the theoretical proofs presented in the paper, particularly Theorem 4.2, which characterizes the score function for the multi-patch data model, and Theorems 4.4 and 4.5, which establish lower bounds on the rule-conforming error (including bias and variance components). Under the stated assumptions, the proofs appear to be mathematically sound and consistent, effectively linking the denoising score matching objective to the inherent constant error in learning fine-grained inter-feature rules.

That said, some aspects rely on idealized conditions (e.g., the use of a simplified two-layer network with linear activation in Theorem 4.5), which may not fully capture the complexities of practical diffusion models. While these simplifications are acceptable for isolating the core theoretical insights, further discussion or empirical validation would help clarify how these bounds translate to more complex architectures encountered in real-world applications.

---

> ### Author Rebuttal · Authors · 2025-04-01
>
> Thanks for your time and efforts reviewing our paper. We now address raised questions as follows.
>
> ---
>
> >Q1: Further ablation studies across different architectures and training configurations.
>
>  ``Section D.3 (Lines 964-1032)`` considers different architectures (U-Net, SiT, DiT) and training configurations including training epochs, training data size and image size. Experimental results show DMs still have limitations in learning fine-grained rules with different settings.
>
> >Q2: The proposed mitigation strategies do not fully resolve the issue.
>
> Thanks for your comments. Our strategy is an initial effort to solve this issue. Importantly, we identify a **potential bottleneck**: the signal of fine rules is too weak to be captured by the classifier, a problem have not been highlighted in classical and Imagenet task-based conditional DDPMs (see ``Section 5.2``). We hope bottleneck analysis can provide valuable insights for future exploration.
>
> Additionally, **the goal of our work** is to reveal the limitations of rule learning in DMs through experiments and theory. Fully addressing this challenge requires further work like rule-specific datasets and metrics, which are currently lacking. We leave the complete resolution to future work.
>
> >Q3:Theoretical proofs are based on simplified models.
>
> Thanks for your question. ``Theorem 4.4`` highlights for non-linear two-layer neural networks, there is constant errors due to the variance of diffusion noise. ``Theorem 4.5`` explicitly derives the constant error. We believe the conclusions from ``Theorem 4.4`` and ``4.5`` extend beyond the simplified setups. Intuitively, the score function is required to satisfy a low-dimensional constraint that holds for every noised input. However, without explicitly embedding this constraint into the model or the training objective, learning it from data becomes inherently difficult. Since the constraint must hold globally, neural networks lack the inductive bias needed to recover such structure from finite samples.
>
> >Q4:Real-world images.
>
> Thank you for your good points. We conduct additional experiments on real-world datasets to further demonstrate DMs can learn coarse rules but struggle with fine ones.
>
> - **SynMirror** [1] presents objects and their reflections, where rules link their features like color, size, and shape. **We find** DDPM captures coarse rules (e.g., matching colors between objects and reflections) but struggles with fine ones, showing shape mismatches.
> - **Cifar-MNIST** pairs specific CIFAR and MNIST classes (e.g., Cats/Dogs with 0/1). **We find** DDPM satisfies coarse rules (e.g., always generating two digits and two objects), but only 20% of generations follow fine-grained rules requiring specific class pairings.
>
> See [Real-world Data](https://anonymous.4open.science/api/repo/Rebuttal-8656/file/exp_ID15070.pdf?v=e6e553d4) for more details. We will add these into revised manuscript.
>
> >Q5:The sensitivity of feature extraction process to hyperparameters (e.g., predefined HSV ranges).
>
> Sorry for the confusion. **There is no hyperparameters in the feature extraction process**. The HSV values used are predefined during training data construction. For example, in Task A, the sun’s HSV is set to yellow with hue [0, 30], saturation [100, 255], and value [200, 255]. The same HSV range is used during feature extraction (see ``Lines 747–806`` for details).
>
> >Q6: Potential future directions.
>
> Inspired by [2,3], one potential direction is optimizing the sampling process during inference. We can introduce additional reward signals from human feedback or reward models to guide DMs during sampling. Additionally, improving the tokenizer to better learn semantic information related to rules could also enhance rule learning. We will add this discussion to revised manuscript.
>
> >Q7: Can you provide more insights into why these approaches yield limited gains?
>
> Thank you for mentioning this question. ``Section 5.2 (Lines 408-424)`` shows limited improvements is due to weak signals of the fine-grained rule which make the guidance from the classifier isn't strong enough to completely correct the sampling. Specially,
> - ``Figure 16``  shows inseparable CLIP representations of contrastive data, making classifier training challenging.
> - ``Figure 17`` demonstrates that training on simple contrastive data results in test accuracy below 90%, highlighting the difficulty in distinguishing subtle differences.
> - [Visualization](https://anonymous.4open.science/api/repo/Rebuttal-8656/file/vis_ID15070.pdf?v=4c7afd58) demonstrates the weak differences between different classes in raw contrastive data.
>
> [1] Reflecting Reality: Enabling Diffusion Models to Produce Faithful Mirror Reflections.
>
> [2] Aligning Text-to-Image Models Using Human Feedback.
>
> [3] Human preference score: Better aligning text-to-image models with human preference.
>
> -----
> We hope above response can address your concern and we are open to discuss more if any question still hold.

---

> > ### Comment · Reviewer_9KzS · 2025-04-05
> >
> > Thank you for the detailed responses, which effectively clarify my questions with detailed ablation studies, real-world experiments, and clear explanations of theoretical and practical limitations. I’m satisfied with these clarifications and will maintain my positive score.

---

> > > ### Author Response · Authors · 2025-04-05
> > >
> > > Dear Reviewer 9KzS,
> > >
> > > We are delighted to hear that our rebuttal has addressed your concerns, and we sincerely appreciate your positive feedback on our work. Thank you for your constructive comments regarding the real-world experiments and additional clarifications. We will include them in the revised manuscript.
> > >
> > > Thank you again for your efforts and time.
> > >
> > > Best,
> > >
> > > Authors

---

### Official Review · Reviewer_KnJN · 2025-03-14

**Overall Recommendation:** 3

**Summary:**

This paper evaluates diffusion models from both experimental and theoretical perspectives on inter-feature rule learning, indicating that while they can capture coarse rules, they struggle with fine-grained ones. The authors also provide a preliminary method to mitigate this shortcoming in learning fine-grained rules.

**Claims And Evidence:**

Yes.

**Essential References Not Discussed:**

None

**Experimental Designs Or Analyses:**

Yes, the experimental designs are reasonable, evaluating the ability of diffusion models to learn physical rules using four carefully designed tasks.

**Methods And Evaluation Criteria:**

The motivation for evaluating the ability of diffusion models to learn fine-grained rules is well-justified and necessary, as it aims to address a major concern that limits the quality of generated outputs in recent large diffusion models.

**Other Comments Or Suggestions:**

None

**Other Strengths And Weaknesses:**

### Strengths
- This paper provides a comprehensive evaluation of whether diffusion models can learn fine-grained inter-feature rules through both experimental and theoretical analysis.

### Weaknesses
- The proposed approach to facilitating fine-grained rule learning appears to have no direct connection with the theoretical analysis and achieves only limited improvements.

**Questions For Authors:**

I think the proposed method to facilitate fine-grained rule learning is a bit straightforward and unrelated to the main analysis of this paper. I hope the authors can clarify this point, and I will adjust my score accordingly.

**Relation To Broader Scientific Literature:**

This paper is closely related to text-to-image generation and highlights a key limitation of recent large diffusion models: their difficulty in learning fine-grained inter-feature rules.

**Theoretical Claims:**

Yes, the proof is logically clear and correct.

---

> ### Author Rebuttal · Authors · 2025-04-01
>
> We thank the reviewer's efforts on reviewing this paper. We now address the questions raised as follows.
>
> ---
>
> >Q1:The proposed approach to facilitating fine-grained rule learning appears to have no direct connection with the theoretical analysis. /The proposed method is unrelated to the main analysis of this paper
>
> Thank you for the question. Our **theoretical analysis** in main text highlights that the failure of classical DDPMs in learning inter-feature rules is mainly due to the objective of denoising, which does not explicitly capture the hidden inter-feature rules. Therefore, classical DDPMs, when trained solely with the standard objective, lack the inductive bias necessary to learn fine-grained inter-feature rules. **This naturally inspires us to introduce additional guidance to steer the sampling process**, encouraging DDPM to generate rule-conforming samples.
>
> Additionally, ``Figure 5`` in **experimental analysis** in main text shows that DMs can generate high-quality samples that meet fine-grained rules, but the process is unstable and prone to rule violations. Therefore, our proposed method introduce additional information to help DDPM stably sample from high-quality regions (more discussion in ``Lines 300–322``).
>
> We will add more discussions in our revised version.
>
> >Q2: The proposed method to facilitate fine-grained rule learning is a bit straightforward/ The proposed approach to facilitating fine-grained rule learning d achieves only limited improvements.
>
> Thank you for your question. The **main focus of our work** is to identify the limitations of DMs in learning fine-grained rules through experiments and theoretical analysis, rather than to propose a complete solution. This limitation has been overlooked (as discussed in ``Section 2``) and represents 'a relatively underexplored area' (as noted by Reviewer 9KzS). Our work aims to 'extend previous findings on compositionality and factual consistency in diffusion models' (as noted by Reviewer gKDf).
>
> Additionally, the proposed method is an initial attempt to enhance rule learning. Importantly, we identify a **key bottleneck**: the signal of fine-grained rules is too weak for the classifier to capture, a phenomenon not been highlighted in traditional DDPMs, such as those targeting ImageNet tasks with classifier guidance (see ``Section 5.2``). We hope that these early attempts and bottleneck analysis can provide valuable insights for future exploration.
>
> [1] Inference-Time Alignment in Diffusion Models with Reward-Guided Generation: Tutorial and Review.
>
> -- ----
>
> We hope the above response resolved the questions and if there is further concern please let us know.

---

> > ### Comment · Reviewer_KnJN · 2025-04-04
> >
> > Thank you for the rebuttal. Some of my concerns have been partially addressed, and I will accordingly raise my score to 3.

---

> > > ### Author Response · Authors · 2025-04-04
> > >
> > > Dear Reviewer KnJN,
> > >
> > > We are glad to hear that our rebuttal has addressed your concerns, and we sincerely appreciate your decision to raise the score to a 3. In particular, thank you for emphasizing the connection between the methodology and the theoretical/experimental sections — we will improve this part in the revised manuscript.
> > >
> > > Thank you again for your effort in reviewing our work.
> > >
> > > Best,
> > >
> > > Authors

---

### Official Review · Reviewer_67RS · 2025-03-14

**Overall Recommendation:** 3

**Summary:**

This paper is motivated by some prevalent real world failure case of diffusion model learning rules between spatial parts and features. They developed a few synthetic tasks to test the learning of diffusion model on inter object rules (spatial or non spatial). Though the overall layout is correct and rough scene rule is obeyed, more precise (linear) spatial relation was kind of obeyed, but not accurate (in the sense of R2\neq1). Then the authors developed a theoretical set up to explain why diffusion training do not lead to rule learning, and proved that with certain theoretical setting (patch data, inter patch rule and separable score function approximator per patch) then the network cannot learn the rule, with a constant error bound on the rule.

Finally the authors developed a few simple yet effective way to mitigate rule confirming problems, e.g. via guided sampling or posthoc rejective sampling.

**Claims And Evidence:**

This paper provided ample evidence for most of its claims, and it’s a well completed paper. I totally agree there are rule learning issues in the practical and synthetic setups. However, I have some issue with the conclusion from theory.

- **Main issue:** I get the theorems in the paper, they are correct. But the overall claim / conclusion made in the abstract that “*Our theoretical analysis demonstrates that DMs trained via denoising score matching (DSM) exhibit constant errors in learning hidden rules, as the DSM objective is not compatible with rule conformity.*” has some issue.
    - This claim is based on a specific theoretical set up where two patches are strongly correlated. We can say the data is supported on a effectively 1d manifold like $\zeta[u; -v]$, with certain offset.
    - However the authors also designed a patch-wise neural network model. Where the output only depends on corresponding part of input! This design made it basically impossible to approximate the true score of the data manifold.
    - Consider an analytically solvable case where $\zeta \sim \mathcal N(0,1)$ then $[x]\sim \mathcal N([0;v],[u;-v][u;-v]^T)$ is distributed as a degenerate Gaussian, with only one nonzero eigenvector in covariance $[u;-v]$. Then its score at any given moment is tractable (Eq 5 [WV2024]).  For Gaussian $\mathcal N(\mu,\sigma^2I+\Sigma)$, the score is a linear function, and it looks like $(\sigma^2I+\Sigma)^{-1}(\mu-x)=\sum_i\frac{1}{\lambda_i+\sigma^2}\nu_i\nu_i^T(\mu-x)$. Basically it’s a full matrix. with a major component spanned by the PC of data $[u;-v][u;-v]^T$. So the score $s^{(1)}$ part depends on both the state $x^{(1)},x^{(2)}$. However your network design prohibit it from depending on the $x^{(2)}$ which **definitely** causes it to be unable to approximate the true score or learn the true data manifold. In the linear case as in Theorem 4.5, the effective weight matrix in your network is block diagonal and each block is rank 1. But the true score for Gaussian requires a full matrix. and the off diagonal blocks cannot be zero.
    - On a higher level, basically you designed a separable network, and the loss is also separable, so basically each patch network learns its own distribution, and it cannot learn the correlation between the two patches. in the end it can only learned a factorized distribution.
    - During rebuttal, the author could address this by modifying the theoretical set up, adding certain setup where dependency is not local to a patch. The authors could also edit their overall claim, and not to attribute the failure of rule learning to the diffusion training on Denoising score matching (DDPM) loss, but to their model design. Currently it seems with a full dependency score model, even being linear function, it will converge to the correct supporting manifold [W2025] (Proposition 5.1). (Though linear network will not learn the correct distribution on manifold, it can only learn Gaussian like things)
- **Minor issue:**
For the empirical experiment on synthetic tasks, whether or not to call it a failure is quite arbitrary. I feel it’s quite successful on ABD. In Figure 5A, the threshold of +-0.01 is quite stringent.

[WV2024] Wang, B., & Vastola, J. J. (2024). The Unreasonable Effectiveness of Gaussian Score Approximation for Diffusion Models and its Applications. TMLR

[W2025] Wang, B. (2025). An Analytical Theory of Power Law Spectral Bias in the Learning Dynamics of Diffusion Models. *arXiv:2503.03206*.

**Essential References Not Discussed:**

One relevant concurrent work that shares the basic theoretical set up with Theorem 4.5 is in **Prop 5.1** in [W2025], i.e. linear symmetric score / denoiser, small or aligned initialization, with slightly more general requirements. Basically their results also confirm that since the data covariance has such low dimensional structure, the weights will automatically discover such low-dimensional structure (feature dimension) by gradient training. But as I pointed out before, in their case the network is overall linear, without adding patch constraint, so their network will recover the correct 1d data manifold as in your case, and will “learn the rule”, or point towards the 1d manifold.

[W2025] Wang, B. (2025). An Analytical Theory of Power Law Spectral Bias in the Learning Dynamics of Diffusion Models. **

**Experimental Designs Or Analyses:**

I applaud the authors for the synthetic data design in this paper, which were well based on actual image diffusion model and their failure cases.

- **Minor issue: interpretation of failure**
For Figure 4, it’s like a half full half empty scenario… whether we call it a success or failure is kind of subjective. To me it’s already quite successful, given A,B,D tasks all have R2~0.80. If you measure correlation you should get sth like 0.90. “…*, where deviations from the ground truth in linear fitting and the coefficient of determination R2 below 1 indicate that DMs fail to fully capture the predefined fine-grained rules*” I feel this is too high a bar to ask for empirical results…
In contrast, for cases in previous works like in Raven’s progression matrices, since the state space is discrete, the evaluation of rule following seems to be more accurate.

**Methods And Evaluation Criteria:**

- I agree that based on current results, one way to enforce better rule conforming is via classifier guidance and post hoc rejection. It's nice that the authors tried this and showed some improvement.

**Other Comments Or Suggestions:**

NA

**Other Strengths And Weaknesses:**

- Overall this is a quite complete paper, showing practical relevance, well motivated set up and theory explaining it, and ways to mitigate the issue. I applaud the authors on such a work!
- The visualizations were well done, and the empirical and theoretical results were stated clearly.

**Questions For Authors:**

- I think the most crucial conceptual question I have is about the distinction about coarse level rule and file grained rules. I can see the distinction in your example, but more generally and theoretically / conceptually, what distinguish coarse level rules from fine grained ones? Why some can be learned and some cannot?
- Similarly can the authors comment on what is spatial rule and what is non spatial rule, or why the task C is not learned as well as A,B,D.,? Does the theory give you any new insights?

**Relation To Broader Scientific Literature:**

The results in Figure 5 B. is quite related to observations in [WSS2024] (Fig.3), for diffusion trained on Raven’s dataset, where the overall rule conforming samples were novel and far from the training set. While local parts of it could be quite similar to some local parts of the dataset. Seems like recombining local parts to create new “scenes”, potentially due to some mechanism as proposed in [KG2024].

[WSS2024] Wang, B., Shang, J., & Sompolinsky, H. (2024). Diverse capability and scaling of diffusion and auto-regressive models when learning abstract rules. NeurIPS Workshop *arXiv:2411.07873*.

[KG2024] Kamb, M., & Ganguli, S. (2024). An analytic theory of creativity in convolutional diffusion models. *arXiv preprint arXiv:2412.20292*.

**Theoretical Claims:**

I checked Theorem 4.2

- At line 299 the statement “*data, requiring that the norm of the first two feature patches sum up to one, i.e., ∥x(1)∥ + ∥x(2)∥ = 1.” **is not correct** in some case**. I think it should be the sum of projection on u an v**  sum to 1.* $\lang u, x^{(1)}\rang+\lang v, x^{(2)}\rang=1$
- See **Claims And Evidence* for a conceptual issue I have with the theoretical treatment in Section 4.**

---

> ### Author Rebuttal · Authors · 2025-04-01
>
> Thanks for your time reviewing this paper. We now address your questions as follows.
>
> ---
> > Q1: Theory
>
> Thanks for your good points.  First, Gaussian setup in [W2025] is a **special case** of our theory. Particularly, as ``Theorem 4.2``, the score can be written as
> $$\nabla \log p_t(x_t^{(1)}, x_t^{(2)}) = - \frac{1}{\beta_t^2} x_t + \frac{\alpha_t}{\beta_t^2} \begin{bmatrix} \gamma(x_t) u , \\ (1- \gamma(x_t)) v \end{bmatrix}$$
> where $\gamma(x_t) = E_{\zeta} [\pi_t (\zeta, x_t) \zeta]$. When $\zeta$ follows Gaussian distribution, the score can be simplified into a linear function over $x_t$, which can be learned by a linear model. However, **we aim to cover more general setup where the above score is *non-linear* in $x_t$**, i.e., $\zeta$ can be any bounded distribution. In such setting, the score function can be generally formulated as a combination of a fixed linear term $-1/\beta_t^2 x_t$ and an additional non-linear term, which motivates us to apply the two-layer network with the residual connection as the score network.
>
> Moreover, our current patch-separated configuration follows prior work (Han et al. 2024a), while similar results can also be extended to network handling dependent patches. Particularly, we can consider
> $$s^{(1,2)}_w(x_t) = - \frac{1}{\beta_t^2} x_t^{(1,2)} + W \sigma(W^\top x_t^{(1,2)})$$
> for some polynomial activation function $\sigma(\cdot)$ and $W \in \mathbb R^{2d \times m}$. Then, it can be seen that only when $\langle W \sigma(W^\top x_t^{(1,2)}), [u; v] \rangle = \alpha_t/\beta_t^2$ for any $x_t$ which concludes the network learns the rule ``(*)``. However, with the new network, we can still follow ``Theorem 4.4`` to show: (1) parameters $W$ will only be the function of $u$, $v$ and the initialization and (2) the network function will be basically a polynomial function of $\langle u, \epsilon_t\rangle$ and $\langle v,\epsilon_t\rangle$ and their cross terms (not appeared in patch-separated network), where $\epsilon_t$ denotes the added diffusion noise in $x_t$. As $x_t$ varies, the function output also varies, which results in a non-vanishing rule conforming error that depends on the variation of $\epsilon_t$. Then similar results in ``Theorem 4.4`` can be obtained and our main theoretical arguments still hold.
>
> > Q2: Interpretation of failure / threshold of Figure 5A / Raven’s matrices
>
> **Interpretation of failure:**
> - Compared to the training data with $R^2$ close to 1 (``Figure 3``), synthetic tasks yield significant lower $R^2$ with 0.6–0.8, indicating weaker rule learning.
> - $R^2$  measures linear fitting quality, not rule accuracy. Differences in coefficients are also important, e.g., Task A’s estimation line $\beta_1 = 0.82$ is smaller than ground truth $\beta_1 = 1$. And ``Error`` metric in ``Table 2`` which combines coefficient deviation and MSE further quantifies rule learning limitation of DMs.
>
> **Figure 5A:** we adopt a strict threshold to show DMs can generate high-quality samples even under such strict conditions. Since DDPM perform well under strict settings, they naturally perform well under more relaxed thresholds.
>
> **Raven Matrices:**  fine-grained rules can also be measured in discrete state space. Specifically, we divide features within [0,1] into 20 intervals. A generation is rule-conforming if measured features fall within the interval satisfies predefined rules. [Discrete Results](https://anonymous.4open.science/api/repo/Rebuttal-8656/file/discretemetric_ID15070.pdf?v=1f32cb35) show 95% of training data satisfy rules, while only 50–80% of generations do, highlighting DMs' limitation in learning rules.
>
> > Q3: line 299 is not correct.
>
> We will modify ``line 299`` to sum of projection.
>
> >Q4: Coarse/fine grained rules.
>
> Fine-grained rules impose stricter requirements than coarse ones. As shown in ``Section 4``, coarse rules only require the network to discover key features $\mathbf{u}$ and $\mathbf{v}$, while fine rules additionally require satisfying constraints between them—e.g., their projections summing to a constant (``Definition 4.3``). Thus, learning coarse rules doesn't guarantee fine-grained rule learning.
>
> >Q5: Spatial/Non-spatial rule.
>
> - Spatial rules involve spatial arrangements like positions and layouts (e.g., Light-shadow in ``Figure 1``), while non-spatial rules relate to features independent of space, such as size and texture in ``Figure 1``. ``Section 3.3`` explains non-spatial rules are harder to learn, possibly due to the lack of explicit cues like positions and lengths in spatial rule, which may explain why Task C is more hard than others.
> - Our theory focuses on spatial rules with clearly defined patch-level dependencies but also can extend to non-spatial rules. For example, by introducing proper tokenizer, inseparable non-spatial rules in pixel space can be separable in latent space, allowing our theory to be applied.
>
> ---
> We hope this addresses your questions. Please let us know if you have any further concerns.

---

> > ### Comment · Reviewer_67RS · 2025-04-07
> >
> > Thank you for the detailed responses and the additional efforts to clarify my questions. I'm pretty happy regarding most of the responses.
> >
> > While most of my concerns have been satisfactorily addressed, I remain unconvinced regarding my first question. I fully agree that the linear function approximator setup in [W2025] is a special case of your formulation, potentially allowing for nonlinearity between the two linear weight matrices. However, my intuition is as follows: if the rule-conforming data reside on a lower-dimensional linear subspace of the two-patch data space—even with a nontrivial distribution in that space—then the optimal linear score (or denoiser), being a linear function of the two patches, should be capable of learning this subspace and achieving perfect rule conformity. Granted, the resulting distribution would be Gaussian for a linear denoiser, and [W2025] suggests that convergence time would be exponentially longer due to small or zero eigenvalues along the tangent subspace. Nonetheless, asymptotically, conformity should be achieved.
> >
> > Given this perspective, I find it difficult to reconcile with your argument:
> >
> > “As  varies, the function output also varies, which results in a non-vanishing rule conforming error that depends on the variation of . Then similar results in Theorem 4.4 can be obtained and our main theoretical arguments still hold.”
> >
> > Could you provide further details of the derivation regarding this point? Although I am pleased with most aspects of the paper, the theoretical argument concerning this issue has not yet convinced me. Based on this concern, I am currently inclined to keep the score as weak reject (2).
> >
> > I look forward to your clarifications on this matter.

---

> > > ### Author Response · Authors · 2025-04-08
> > >
> > > We are glad that our rebuttal has addressed most of your other concerns. Thank you once again for your thoughtful follow-up questions. We would like to take this chance to further clarify the rule-conforming error, especially when there exists a mismatch between the model class and the underlying rule.
> > >
> > > We fully agree with your intuition: if the rule-conforming data lies on low-dimensional *linear* subspace and if the score network is a *linear* function, then the rule-conforming error can vanish asymptotically. This can be also reflected in our ``Theorem 4.4`` (considering using linear network for all patches), where the polynomial functions $\tilde \sigma^{(1)}(\cdot)$ and $\tilde \sigma^{(2)}(\cdot)$, which have polynomial degree 1 (as we consider linear model), will become a constant function, leading to a zero lower bound of the rule-conforming error.
> > >
> > > However, our theoretical argument mainly focuses on the general case, where we consider the setting that
> > > * the underlying rule is *unknown* to the learner, and
> > > * the model class may *not align* with the true structure of the rule, i.e., the score network can be much more complicated than the linear function.
> > >
> > > In such scenarios, the more complicated neural network model will be more powerful to recover the entire data distribution (which could be complicated for non-Gaussian $\zeta$), while the hidden rule may not be well captured. That said, the polynomial functions $\tilde \sigma^{(1)}(\cdot)$ and $\tilde \sigma^{(2)}(\cdot)$ in ``Theorem 4.4`` will be non-constant, Then, the rule-conforming error will be non-zero.
> > >
> > > To provide some theoretical intuitions, we can consider a simple case that the first two patches are $\zeta u$ and $-\zeta v$. In the **linear model setting**, considering $\zeta\sim N(0,1)$, the rule conforming function $\psi_t(x)$ can be roughly written as $\psi_t(x)=\langle f(\Sigma)\Sigma x,[u,v]\rangle$, where $\Sigma$ is the covariance matrix of the data and $f(\Sigma)$ is a function of $\Sigma$ that is commutable with $\Sigma$. Importantly, in this setting, the reason why linear model can handle the linear rule is that the vector $[u,v]$ is just the eigen-vector of $\Sigma$ with eigenvalue $0$, then clearly $\Sigma f(\Sigma) \cdot [u,v]=0$, and thus $\psi_t(x)=0$ for all $x$. However, if using **non-linear models**, for instance, we consider $s_w(x)=W_1x + W_2(x\circ x)$ as an example ($x\circ x$ denotes the hadamard product). Then, we need to consider the covariance matrix over the transformed data $[x, x\circ x]$, which no longer align with the vector $[u,v]$ for general non-Gaussian $\zeta$. As a consequence, as long as $W_2$ is non-zero, we will not be able to obtain $W_2\cdot[u,v]=0$ (as $[u,v]$ will not be in the null space of $W_2$ as in $\Sigma$). Then, we can follow the similar analysis in ``Theorem 4.4`` and show that the conforming function $\psi_t(x)=\langle W_1 x+W_2(x\circ x)+1/\beta_t^2 x,[u,v]\rangle =0$ **will not hold for all $x$**, implying that the rule conforming error will not be zero.
> > >
> > > To further support this claim empirically, we have included an [additional experiment](https://anonymous.4open.science/api/repo/Rebuttal-DDBA/file/diff_0.8_asym.pdf?v=fd07d01b) comparing rule-conforming errors across different model classes in our synthetic *linear* data setup. As shown in the figure, *linear model* achieves significantly lower rule-conforming error compared to more complex, nonlinear models (2-layer, 3-layer MLPs with ReLU or quadratic activation, operated on all patches jointly). This aligns with our claims, i.e., without exact structural alignment between the model/objective and the rule, the small rule-conforming error cannot be guaranteed.
> > >
> > > We hope this explanation clarifies your confusion and we are happy to answer any further questions. We will make sure to include the additional discussions and experiments in our revised version based on your comments. Your feedback has been invaluable in helping us improve the clarity and depth of our paper.

---

### Decision · Program_Chairs · 2025-05-01

**Decision:**

Accept (poster)

**Comment:**

Motivated by real-world failures in how diffusion models learn inter-object rules, this paper introduces synthetic tasks to evaluate spatial and non-spatial reasoning. While models capture rough layouts, precise spatial relations remain inaccurate. The authors develop a theoretical framework showing that, under certain conditions (e.g., patch-based data and separable score approximators), diffusion models are provably unable to learn such rules, incurring a constant error bound.

All reviewers find the result interesting and solid theoretical studies. Please incorporate the reviewers' feedback for revision in camera ready.